# JAWS: Auditing Predictive Uncertainty Under Covariate Shift

**Drew Prinster**
Department of Computer Science
Johns Hopkins University
Baltimore, MD 21211
drew@cs.jhu.edu

**Anqi Liu**
Department of Computer Science
Johns Hopkins University
Baltimore, MD 21211
aliu@cs.jhu.edu

**Suchi Saria**
Department of Computer Science
Johns Hopkins University
Baltimore, MD 21211
ssaria@cs.jhu.edu

## Abstract

We propose **JAWS**, a series of wrapper methods for distribution-free uncertainty quantification tasks under covariate shift, centered on the core method **JAW**, the **JA**ckknife+ **W**eighted with data-dependent likelihood-ratio weights. JAWS also includes computationally efficient **A**pproximations of JAW using higher-order influence functions: **JAWA**. Theoretically, we show that JAW relaxes the jackknife+'s assumption of data exchangeability to achieve the same finite-sample coverage guarantee even under covariate shift. JAWA further approaches the JAW guarantee in the limit of the sample size or the influence function order under common regularity assumptions. Moreover, we propose a general approach to repurposing predictive interval-generating methods and their guarantees to the reverse task: estimating the probability that a prediction is erroneous, based on user-specified error criteria such as a safe or acceptable tolerance threshold around the true label. We then propose **JAW-E** and **JAWA-E** as the repurposed proposed methods for this **E**rror assessment task. Practically, JAWS outperform state-of-the-art predictive inference baselines in a variety of biased real world data sets for interval-generation and error-assessment predictive uncertainty auditing tasks.

## 1 Introduction

**Auditing the uncertainty under data shift** Principled quantification of predictive uncertainty is crucial for enabling users to calibrate how much they should or should not trust a given prediction [Thiebes et al., 2021, Ghosh et al., 2021, Tomsett et al., 2020, Bhatt et al., 2021]. Uncertainty-based predictor auditing can be considered a type of uncertainty quantification performed *post-hoc*, for example by a regulator without detailed knowledge of a predictor's architecture and with limited resources [Schulam and Saria, 2019]. Data shift poses a major challenge to uncertainty quantification due to violation of the common assumption that the training and test data are exchangeable, or more specifically independent and identically distributed (i.i.d.) [Ovadia et al., 2019, Ulmer et al., 2020, Zhou and Levine, 2021, Chan et al., 2020]. Therefore, it is essential to develop convenient tools for users or regulators to audit the uncertainty of a given prediction even when training data is biased.

**Predictor auditing: Interval generation** In this work we distinguish between two types of predictive uncertainty auditing. We describe the first type as *interval generation*, which refers to a common

goal in the distribution-free uncertainty quantification literature: to generate a predictive confidence interval (or set) that covers the true label with at least a user-specified probability. For instance, an auditor might ask for predictive intervals that contain the true label with at least, say 90% frequency.

**Predictor auditing: Error assessment** While predictive interval generation has been a central focus of the distribution-free uncertainty quantification literature [Angelopoulos and Bates, 2021], in some applications the reverse computation may be more actionable: estimating the probability that a prediction is erroneous or not, based on user-specified error critieria such as a safe or acceptable tolerance region around the true label. We thus refer to this task as *error assessment*. For instance, take the setting of chemical or radiation therapy dose prediction for cancer treatment, where administering a dose within approximately $\pm 10\%$ of the optimal dose is considered safety-critical (see Appendix A.1 for details). Whereas predictive interval generation could fail to provide safety assurance (e.g., if the predictive confidence interval is larger than the safe tolerance region), error assessment would give a worst-case probability of the prediction being safe. Similar examples could be formulated in other applications, such as incision planning in surgical robotics and autonomous vehicle navigation.

**Coverage** We assume a standard regression setup with a multiset of training data $\{(X_1, Y_1), ..., (X_n, Y_n)\}$ and a test point $(X_{n+1}, Y_{n+1})$ with unknown label $Y_{n+1}$, where $(X_i, Y_i) \in \mathbb{R}^d \times \mathbb{R}$ for all $i \in \{1, ..., n+1\}$. Also, we denote a predictor as $\widehat{\mu} = \mathcal{A}(\{(X_1, Y_1), ..., (X_n, Y_n)\})$, where $\mathcal{A}$ is a model-fitting algorithm. For a predictive interval (or set) $\widehat{C}_{n,\alpha}^{\text{audit}} : \mathbb{R}^d \rightarrow \{\text{subsets of } \mathbb{R}\}$, a *coverage guarantee* gives a lower bound to the probability that the interval covers the true test label:

$$\mathbb{P}\{Y_{n+1} \in \widehat{C}_{n,\alpha}^{\text{audit}}(X_{n+1})\} \geq 1 - \alpha. \tag{1}$$

The coverage guarantee provides the basis for both interval-generation and error-assessment auditing, though it is important to note that in this work we focus on marginal rather than conditional coverage (see [Foygel Barber et al., 2021] for more details on this distinction). Standard conformal prediction methods [Vovk et al., 2005, Shafer and Vovk, 2008, Vovk, 2013] along with the jackknife+ and related methods [Barber et al., 2021], which we refer to together as "predictive inference" methods, provide a framework for generating predictive intervals with finite-sample guaranteed coverage.

**Exchangeability** Standard conformal prediction and the jackknife+ rely on two crucial notions of *exchangeability*: data exchangeability, that is that the training and test data are all exchangeable (e.g., i.i.d.); and secondly that the model-fitting algorithm $\mathcal{A}$ treats the data symmetrically [Barber et al., 2022]. In common situations of dataset shift, however, the data exchangeability assumption is violated. Empirically, the coverage performance of standard conformal prediction methods can suffer under data shift [Tibshirani et al., 2019, Podkopaev and Ramdas, 2021].

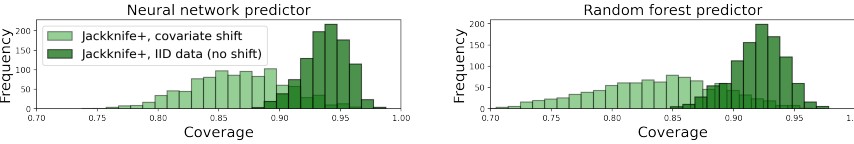

Figure 1: Jackknife+ loses coverage on the airfoil dataset under covariate shift (details in Section 4).

In this work, we build on the jackknife+ method due to its beneficial compromise between the statistical and computational limitations of other conformal prediction methods [Barber et al., 2021]. However, jackknife+ coverage performance can still degrade under data shift, such as shown in Figure 1, and in some applications its computational requirements can still be limiting. To address these concerns and make extensions to error assessment, we develop JAWS, a series of wrapper methods for distribution-free uncertainty quantification under covariate shift (see Table 1 for key properties).

Table 1: Summary of key properties for JAWS methods (details in Section 3).

| Method | Task | Guarantee (under covariate shift) | | Avoids retraining |
|---|---|---|---|---|
| | | **Finite sample** | **Asymptotic** | |
| JAW | Interval generation | ✓ | ✓ | ✗ |
| JAWA | Interval generation | ✗ | ✓ | ✓ |
| JAW-E | Error assessment | ✓ | ✓ | ✗ |
| JAWA-E | Error assessment | ✗ | ✓ | ✓ |

**Our contributions can be summarized as follows**:

1. We develop JAW: a jackknife+ method with data-dependent likelihood-ratio weights for predictive interval generation under covariate shift. We show that JAW achieves the same rigorous, finite-sample coverage guarantee as jackknife+ [Barber et al., 2021] while relaxing the data exchangeability assumption to allow for covariate shift.

2. We develop JAWA: a sequence of computationally efficient approximations to JAW that uses higher-order influence functions to avoid retraining. Under assumptions outlined in Giordano et al. [2019a] regarding the regularity of the data, Hessian of the objective (local strong convexity), and the existence and boundedness of higher order derivatives, we provide an asymptotic guarantee for the JAWA coverage in the limit of the sample size or influence function order.

3. We propose a general approach to repurposing any distribution-free predictive inference method to the error assessment task, with rigorous guarantees for the coverage probability estimation. Our approach applies to methods that assume exchangeable data and to methods like JAW and JAWA that allow for covariate shift—JAW-E and JAWA-E refer to the error assessment versions.

4. We demonstrate superior empirical performance of JAWS over other distribution-free predictive inference baselines on a variety of benchmark datasets under covariate shift.

## 2 Background and related work

### 2.1 Standard conformal prediction

Conformal prediction has grown into a broad research field since arising in the 1990s [Vovk et al., 2005, Shafer and Vovk, 2008, Balasubramanian et al., 2014, Angelopoulos and Bates, 2021]. Standard conformal prediction methods generate a prediction interval (or set) with a finite-sample coverage guarantee as in (1), which is *distribution-free* in the sense that the guarantee applies to any exchangeable data distribution [Lei and Wasserman, 2014, Lei et al., 2018]. With the exchangeability assumptions in Section 1, standard conformal prediction methods rely on a pre-fit score function $\widehat{S} : \mathbb{R}^d \times \mathbb{R} \to \mathbb{R}$ (in regression, the absolute-value residual score $\widehat{S}(x, y) = |y - \widehat{\mu}(x)|$ is commonly used). A conformal prediction interval at confidence level $1 - \alpha$ is then determined by a corresponding quantile on a multiset of (exchangeable) score values.

Split conformal and full conformal are two main types of standard conformal prediction, and each bears its own limitation [Vovk et al., 2005, Shafer and Vovk, 2008]. Split conformal generates scores on labeled holdout data and is computationally efficient due to not requiring retraining, but sample splitting to obtain the holdout set can reduce model accuracy [Papadopoulos, 2008, Lei et al., 2018, Vovk, 2012]. On the other hand, full conformal prediction avoids the holdout set requirement, but at the heavy computational cost of retraining the model on every possible target value (or, in practice, on a fine grid of target values) [Ndiaye and Takeuchi, 2019, Zeni et al., 2020].

### 2.2 Covariate shift

Under the *covariate shift* assumption, the $Y|X$ distribution is assumed to be the same between training and test data but the marginal $X$ distributions may change [Sugiyama et al., 2007, Shimodaira, 2000]:

$$(X_i, Y_i) \overset{\text{i.i.d.}}{\sim} P_X \times P_{Y|X}, i = 1, ..., n; \qquad (X_{n+1}, Y_{n+1}) \sim \widetilde{P}_X \times P_{Y|X}, \text{ independently.} \quad (2)$$

Rich literature exist in this domain—see Appendix A.2 for more details. Uncertainty quantification is relatively less explored under covariate shift, though recent work [Ovadia et al., 2019, Zhou and Levine, 2021, Chan et al., 2020] emphasizes its importance, especially in deep learning.

### 2.3 Conformal prediction under covariate shift and beyond exchangeability

Tibshirani et al. [2019] develop the idea of *weighted exchangeability* for adapting conformal prediction to the covariate shift setting. Random variables $V_1, ..., V_n$ are weighted exchangeable with weight functions $w_1, ..., w_n$ if their joint density $f$ can be factorized as $f(v_1, ..., v_n) = \prod_{i=1}^{n} w_i(v_i) \cdot g(v_1, ..., v_n)$, where $g$ is independent of ordering on its inputs. For covariate shift as in (2), if $\widetilde{P}_X$ is absolutely continuous with respect to $P_X$, then the data $\{(X_i, Y_i)\}$ are weighted exchangeable with weight functions $w_1 = ... = w_n = 1$ and $w_{n+1} = w = d\widetilde{P}_X / dP_X$ [Tibshirani et al., 2019].

If $\{v_i\}$ represents a set of scores for standard conformal prediction, then we can represent the empirical distribution of $\{v_i\}$ as $\frac{1}{n+1}\sum_{i=1}^{n}\delta_{v_i} + \frac{1}{n+1}\delta_\infty$, where $\delta_{v_i}$ denotes a point mass at $v_i$ [Barber et al., 2022]. By extension, weighted conformal prediction uses the *weighted* empirical distribution defined as $\sum_{i=1}^{n} p_i^w(x)\delta_{v_i} + p_{n+1}^w(x)\delta_\infty$, with weights given by

$$p_i^w(x) = \frac{w(X_i)}{\sum_{j=1}^{n} w(X_j) + w(x)}, i = 1, ..., n; \quad \text{and} \quad p_{n+1}^w(x) = \frac{w(x)}{\sum_{j=1}^{n} w(X_j) + w(x)}, \quad (3)$$

where $w = \mathrm{d}\widetilde{P}_X/\mathrm{d}P_X$, so $p_i^w(X_{n+1})$ can be thought of as a normalized likelihood ratio weight for each $i \in \{1, ..., n+1\}$. Corollary 1 in [Tibshirani et al., 2019] provides the coverage guarantee of weighted conformal prediction that takes the form of (1) but relaxes the exchangeable data assumption to allow for covariate shift. However, weighted split and weighted full conformal inherit the same statistical and computational limitations, respectively, from their standard (exchangeable) variants.

The recent work of [Barber et al., 2022] provides a novel extension of conformal prediction and the jackknife+ to unknown violations of the exchangeability assumption, including a "nonexchangeable jackknife+" defined with fixed weights. The key difference between the nonexchangeable jackknife+ in Barber et al. [2022] and our proposed JAW method is that Barber et al. [2022] use fixed weights to compensate for unknown exchangeability violations (not limited to covariate shift) but at the expense of a bounded but generally nonzero "coverage gap" (drop in guaranteed coverage relative to if the data were exchangeable), whereas our JAW method with data-dependent weights assumes covariate shift but does not suffer from any similar coverage gap. See Appendix A.3 for more details.

## 2.4 Jackknife+

The jackknife+ [Barber et al., 2021], which is closely related to cross conformal prediction [Vovk et al., 2018], offers a compromise between the statistical limitation of split conformal and the computational limitation of full conformal, at the cost of a slightly weaker coverage guarantee. The jackknife+ predictive interval can most easily be understood as a modification to a predictive interval from the classic jackknife resampling method [Miller, 1974, Steinberger and Leeb, 2018, 2016]. For a set of point masses $\{\delta_{v_i}\}$ at values $v_1, ..., v_n$, let $Q_\beta^-\{\frac{1}{n+1}\delta_{v_i}\}$ denote the level $\beta$ quantile on the empirical distribution $\sum_{i=1}^{n}\frac{1}{n+1}\delta_{v_i} + \frac{1}{n+1}\delta_{-\infty}$ and let $Q_\beta^+\{\frac{1}{n+1}\delta_{v_i}\}$ denote the level $\beta$ quantile on the empirical distribution $\sum_{i=1}^{n}\frac{1}{n+1}\delta_{v_i} + \frac{1}{n+1}\delta_\infty$. Then, denoting the model trained without the $i$th point as $\widehat{\mu}_{-i} = \mathcal{A}\big(\{(X_1, Y_1), ..., (X_{i-1}, Y_{i-1}), (X_{i+1}, Y_{i+1}), ..., (X_n, Y_n)\}\big)$ and the leave-one-out residual $R_i^{LOO} = |Y_i - \widehat{\mu}_{-i}(X_i)|$, the jackknife prediction interval can be written as

$$\widehat{C}_{n,\alpha}^{\text{jackknife}}(X_{n+1}) = \left[ Q_\alpha^-\left\{ \tfrac{1}{n+1}\delta_{\widehat{\mu}(X_{n+1}) - R_i^{LOO}} \right\}, Q_{1-\alpha}^+\left\{ \tfrac{1}{n+1}\delta_{\widehat{\mu}(X_{n+1}) + R_i^{LOO}} \right\} \right]. \quad (4)$$

In contrast, we obtain the jackknife+ predictive interval in Barber et al. [2021] by replacing the full model prediction $\widehat{\mu}(X_{n+1})$ in (4) with $\widehat{\mu}_{-i}(X_{n+1})$:

$$\widehat{C}_{n,\alpha}^{\text{jackknife+}}(X_{n+1}) = \left[ Q_\alpha^-\left\{ \tfrac{1}{n+1}\delta_{\widehat{\mu}_{-i}(X_{n+1}) - R_i^{LOO}} \right\}, Q_{1-\alpha}^+\left\{ \tfrac{1}{n+1}\delta_{\widehat{\mu}_{-i}(X_{n+1}) + R_i^{LOO}} \right\} \right]. \quad (5)$$

[Barber et al., 2021] prove that, with the same exchangeability assumptions as in standard conformal prediction, the jackknife+ prediction interval satisfies

$$\mathbb{P}\{Y_{n+1} \in \widehat{C}_{n,\alpha}^{\text{jackknife+}}(X_{n+1})\} \geq 1 - 2\alpha. \quad (6)$$

## 2.5 Approximating leave-one-out models with higher-order influence functions

Influence functions (IFs) [Cook, 1977] have a long history in robust statistics for estimating the dependence of parameters on sample data. Recently, IFs have become more widespread in machine learning for uses including model interpretability Koh and Liang [2017] and approximating classic resampling-based uncertainty quantification methods including bootstrap [Schulam and Saria, 2019], jackknife, and leave-$k$-out cross validation [Giordano et al., 2019b,a]. In each of these cases, IFs enable approximation of the parameters that would be obtained if the model were retrained on resampled data by instead estimating the effect of a corresponding reweighting. In prior work, Alaa and Van Der Schaar [2020] proposed approximating the leave-one-out models required by the jackknife+ with higher-order IFs, but their work assumes exchangeable or i.i.d. train and test data.

Let $\hat{\theta}$ denote the fitted parameters for predictor $\widehat{\mu}$ trained on the full training data. Given Assumptions 1-4 in Giordano et al. [2019a]—which require that $\hat{\theta}$ is a local minimum of the objective function, that the objective is $k+1$ times continuously differentiable with bounded norms, and that the objective is strongly convex in the neighborhood of $\hat{\theta}$—then the $k$-th order leave-one-out IF refers to the $k$-th order directional derivative of the model parameters $\hat{\theta}$ with respect to the data weights, in the direction of the leave-one-out change in weights (See A.4 for more details). With each of these $k$th order leave-one-out IFs for $k \in \{1, ..., K\}$, denoted with condensed notation $\delta_{-i}^k \hat{\theta}$, we can construct a $K$-th order Taylor series approximation to estimated the leave-one-out model parameters $\hat{\theta}_{-i}$ :

$$\hat{\theta}_{-i}^{\text{IF-}K} := \hat{\theta} + \sum_{k=1}^{K} \frac{1}{k!} \delta_{-i}^k \hat{\theta}. \tag{7}$$

In this work we implement the algorithm proposed by Giordano et al. [2019a] to compute higher-order IFs, a recursive procedure based on foreward-mode automatic differentiation [Maclaurin et al., 2015] for memory efficiency in computing higher-order directional derivatives. Our introduction of IFs is highly simplified—we refer to Appendix A.4 and to Giordano et al. [2019a] for more details.

## 2.6 Error assessment

Whereas conformal prediction and related methods generate prediction intervals that control the error probability (miscoverage level $\alpha$) at a user-specified level, we refer to the reverse task as *error assessment*: estimating the probability that a prediction is erroneous or not, based on user-specified error criteria. For instance, a user might define an error as any deviation between the prediction $\widehat{\mu}(X_{n+1})$ and the true label $Y_{n+1}$ greater than some acceptable tolerance threshold $\tau$: that is, when $|Y_{n+1} - \widehat{\mu}(X_{n+1})| > \tau$. In Section 3.3, we present a general approach to repurposing predictive inference methods with validity under covariate shift to error assessment.

We note that for score functions that are monotonic in $y$, such as $\widehat{S}(x, y) = y - \widehat{\mu}(x)$, guarantees for this error assessment task can be obtained using conformal predictive *distributions* as described by Vovk et al. [2017] (also see Vovk et al. [2020], Vovk and Bendtsen [2018], Xie and Zheng [2022]). In regression tasks assuming exchangeable data, CPDs generate a probability distribution for the label over $\mathbb{R}$. However, CPDs require that score functions be monotonic in $y$, whereas we allow for certain non-monotone conformity scores such as the commonly used absolute-value residual $|y - \widehat{\mu}(x)|$; Moreover, CPDs assume exchangeable data, whereas our approach extends to covariate shift.

# 3 Proposed approach and theoretical results

## 3.1 JAW: Jackknife+ weighted with data-dependent weights

We present **JAW**, the **JA**ckknife+ **W**eighted with data-dependent likelihood-ratio weights, defined by the following predictive interval:

$$\widehat{C}_{n,\alpha}^{\text{JAW}}(X_{n+1}) = \left[ Q_{\alpha}^{-} \left\{ p_i^w(X_{n+1}) \cdot \delta_{\widehat{\mu}_{-i}(X_{n+1}) - R_i^{LOO}} \right\}, \; Q_{1-\alpha}^{+} \left\{ p_i^w(X_{n+1}) \cdot \delta_{\widehat{\mu}_{-i}(X_{n+1}) + R_i^{LOO}} \right\} \right], \tag{8}$$

where $R_i^{LOO} = |\widehat{\mu}_{-i}(X_i) - Y_i|$, with $p_i^w(x)$ for $i \in \{1, ..., n+1\}$ as in (3), where $Q_{\alpha}^{-} \{ p_i^w(X_{n+1}) \cdot \delta_{\widehat{\mu}_{-i}(X_{n+1}) - R_i^{LOO}} \}$ denotes the level $\alpha$ quantile of the empirical distribution $\sum_{i=1}^{n} \left[ p_i^w(X_{n+1}) \cdot \delta_{\widehat{\mu}_{-i}(X_{n+1}) - R_i^{LOO}} \right] + p_{n+1}^w(X_{n+1}) \cdot \delta_{-\infty}$, and where $Q_{1-\alpha}^{+} \{ p_i^w(X_{n+1}) \cdot \delta_{\widehat{\mu}_{-i}(X_{n+1}) + R_i^{LOO}} \}$ is the level $1-\alpha$ quantile for $\sum_{i=1}^{n} \left[ p_i^w(X_{n+1}) \cdot \delta_{\widehat{\mu}_{-i}(X_{n+1}) + R_i^{LOO}} \right] + p_{n+1}^w(X_{n+1}) \cdot \delta_{\infty}$.

We choose to define JAW using likelihood-ratio weights $w(X_i) = d\widetilde{P}_X(X_i)/dP_X(X_i)$ in the $p_i^w(x)$ to address covariate shift, but a similar result holds for other instances of weighted exchangeability and corresponding data-dependent weight functions (see Appendix B.1). We show that $\widehat{C}_{n,\alpha}^{\text{JAW}}(X_{n+1})$ satisfies the same coverage guarantee as the jackknife+ except relaxing the data exchangeability assumption to allow for covariate shift, which we state formally in the following theorem.

**Theorem 1.** *Assume data under covariate shift from* (2). *If* $\widetilde{P}_X$ *is absolutely continuous with respect to* $P_X$, *then the JAW interval in* (8) *satisfies*

$$\mathbb{P}\{Y_{n+1} \in \widehat{C}_{n,\alpha}^{JAW}(X_{n+1})\} \geq 1 - 2\alpha. \tag{9}$$

**Remark 1.** The results from Tibshirani et al. [2019] do not directly imply Theorem 1. The approach in Tibshirani et al. [2019] relies on leveraging the weighted exchangeability of the data to reweight the nonconformity scores $\{V_1, ..., V_{n+1}\}$ so they can be treated as exchangeable, and for the jackknife+ this approach would entail treating $\widehat{\mu}_{-i}(X_{n+1}) \pm R_i^{LOO}$ as implicit nonconformity scores. But, observe that for $i \in \{1, ..., n\}$, $\widehat{\mu}_{-i}$ is trained on $n - 1$ datapoints, whereas $\widehat{\mu}_{-(n+1)} = \widehat{\mu}$ is trained on $n$ datapoints. Thus, no reweighting can make $\widehat{\mu}_{-i}$ equivalent in distribution to $\widehat{\mu}$ and thereby allow us to treat the reweighted $\widehat{\mu}_{-i}(X_{n+1}) \pm R_i^{LOO}$ and $\widehat{\mu}(X_{n+1}) \pm R_{n+1}^{LOO}$ as exchangeable.

**Proof sketch:** Our proof technique for Theorem 1 extends the jackknife+ coverage guarantee proof in Barber et al. [2021] to the covariate shift setting for JAW using likelihood ratio weights as in Tibshirani et al. [2019]. The full proof is given in Appendix C.1, but the outline is as follows:

*Setup*: Following Barber et al. [2021], we define a set of leave-*two*-out models $\{\tilde{\mu}_{-(i,j)}\}$. We then generalize the notion of "strange" points described in Barber et al. [2021] to covariate shift.

1. *Bounding the total normalized weight of strange points*: We establish deterministically that the total normalized weight of strange points cannot exceed $2\alpha$.

2. *Weighted exchangeability using the leave-two-out models*: Using the leave-two-out model construction, we leverage weighted exchangeability to show that the probability that a test point $n + 1$ is strange is thus bounded by $2\alpha$.

3. *Connection to JAW*: Lastly, we show that the JAW interval can only fail to cover the test label value $Y_{n+1}$ if $n + 1$ is a strange point.

While JAW assumes access to oracle likelihood ratio weights, in practice this information often has to be estimated. See Appendix D.5 for a discussion and experiments of JAW with estimated weights.

### 3.2 JAWA: Using higher-order influence functions to approximate JAW without retraining

For computationally efficient JAW **A**pproximations that avoid retraining $n$ leave-one-out models, we propose the JAWA sequence, which approximates the leave-one-out models required by JAW using higher-order influence functions. For each training point $i \in \{1, ..., n\}$, define the $K$-th order influence function approximation to the leave-one-out refit parameters $\hat{\theta}_{-i}$, obtained from Algorithm 4 in Giordano et al. [2019a], as given by equation (7), and let $\widehat{\mu}_{-i}^{\text{IF-}K}$ be the model with with these approximated parameters $\hat{\theta}_{-i}^{\text{IF-}K}$ for each $i \in \{1, ..., n\}$. Then, the prediction interval for the $K$-th order JAWA (i.e., for JAWA-$K$) is given by

$$\widehat{C}_{n,\alpha}^{\text{JAWA-}K}(X_{n+1}) = \Big[ Q_\alpha^- \big\{ p_i^w(X_{n+1}) \cdot \delta_{\widehat{\mu}_{-i}^{\text{IF-}K}(X_{n+1}) - R_i^{\text{IF-}K,LOO}} \big\},$$
$$Q_{1-\alpha}^+ \big\{ p_i^w(X_{n+1}) \cdot \delta_{\widehat{\mu}_{-i}^{\text{IF-}K}(X_{n+1}) + R_i^{\text{IF-}K,LOO}} \big\} \Big], \qquad (10)$$

with $R_i^{\text{IF-}K,LOO} = \big| \widehat{\mu}_{-i}^{\text{IF-}K}(X_i) - Y_i \big|$, $p_i^w(x)$ as in (3), and quantiles defined analogously to JAW.

We now provide an asymptotic coverage guarantee for $\widehat{C}_{n,\alpha}^{\text{JAWA-}K}(X_{n+1})$ that holds either in the limit of the sample size or in the limit of the influence function order, under regularity conditions formally described in Giordano et al. [2019a]. These assumptions concern the regularity and continuity of the training data, local convexity of the objective (or that the Hessian is strongly positive definite), and the existence and boundedness of the objective's 1st through $K + 1$th order directional derivatives.

**Theorem 2.** *Assume data under covariate shift from (2) and that $\widetilde{P}_X$ is absolutely continuous with respect to $P_X$. Let Assumptions 1 - 4 and either Condition 2 or Condition 4 from Giordano et al. [2019a] hold uniformly for all $n$. Then, in the limit of the training sample size $n \to \infty$ or in the limit of the influence function order $K \to \infty$, the JAWA-$K$ interval in (10) satisfies*

$$\mathbb{P}\big\{ Y_{n+1} \in \widehat{C}_{n,\alpha}^{\text{JAWA-}K}(X_{n+1}) \big\} \geq 1 - 2\alpha \qquad (11)$$

We leave the proof to Appendix C.2, but we note that the result follows by combining Propositions 1 and 3 in Giordano et al. [2019a] with the JAW coverage guarantee that we present in Theorem 1.

### 3.3 Error assessment under covariate shift

We now propose a general approach to repurposing predictive inference methods with validity under covariate shift from predictive interval generation to the reverse task: estimating the probability that a

prediction is erroneous or not, based on user-specified error criteria. For example, consider a user that defines a prediction $\widehat{\mu}(X_{n+1})$ as erroneous, relative to the true label $Y_{n+1}$, if it is farther than some acceptable tolerance threshold $\tau$ from $Y_{n+1}$: i.e., if $|Y_{n+1} - \widehat{\mu}(X_{n+1})| > \tau$. For this common regression error criterion, our approach to adapting a method such as JAW (8) or weighted split conformal prediction [Tibshirani et al., 2019] to error assessment reduces to first defining the set of labels that would *not* be considered erroneous, $\overline{E} = [\widehat{\mu}(X_{n+1}) - \tau, \ \widehat{\mu}(X_{n+1}) + \tau]$, and then finding the method's largest predictive interval contained within $\overline{E}$, call it $\widehat{C}_{n,\alpha_E}^{\text{w-audit}}(X_{n+1})$. The coverage guarantee for $\widehat{C}_{n,\alpha_E}^{\text{w-audit}}(X_{n+1})$ then yields a lower bound on $\mathbb{P}\{Y_{n+1} \in \overline{E}\}$, the probability of no error (or an upper bound on the error probability). See Figure 2 for an illustration of this example.

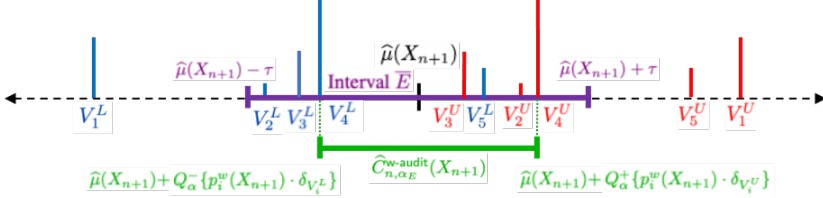

Figure 2: Illustration of approach to repurposing a predictive inference method "w-audit" to error assessment. The interval $\overline{E} = [\widehat{\mu}(X_{n+1}) - \tau, \ \widehat{\mu}(X_{n+1}) + \tau]$ is shown in violet, the lower score values $\{V_i^L\}$ in blue, the upper score values $\{V_i^U\}$ in red, and the interval $\widehat{C}_{n,\alpha_E}^{\text{w-audit}}(X_{n+1})$ in green. Each vertical line at a location $V_i$ on the real line represents a point mass $\delta_{V_i}$ with height corresponding to the normalized likelihood ratio weight $p_i^w(X_{n+1})$.

Generally, a user must specify error criteria by a test point score function $\widehat{S} : \mathbb{R}^d \times \mathbb{R} \to \mathbb{R}$ (for conformal prediction, a nonconformity score), as well as minimum and maximum acceptable score values $\tau^-$ and $\tau^+$, where $\tau^- < \tau^+$; i.e., $\widehat{\mu}(X_{n+1})$ is considered erroneous if $\widehat{S}(X_{n+1}, Y_{n+1}) < \tau^-$ or if $\tau^+ < \widehat{S}(X_{n+1}, Y_{n+1})$. (For nonnegative $\widehat{S}$, we might let $\tau^- = 0$.) Then, the values of $y$ for which observing $Y_{n+1} = y$ would *not* imply $\widehat{\mu}(X_{n+1})$ is erroneous are:

$$\overline{E} = \{y \in \mathbb{R} : \tau^- \le \widehat{S}(X_{n+1}, y) \le \tau^+\}. \tag{12}$$

Now, assume a predictive inference method with predictive sets that can be written in the form

$$\widehat{C}_{n,\alpha}^{\text{w-audit}}(X_{n+1}) = \left\{ y \in \mathbb{R} : \widehat{Q}_\alpha^- \{p_i^w(X_{n+1})\delta_{V_i^L}\} \le \widehat{S}(X_{n+1}, y) \le \widehat{Q}_{1-\alpha}^+ \{p_i^w(X_{n+1})\delta_{V_i^U}\} \right\} \tag{13}$$

with valid coverage guaranteed under covariate shift. (Note, (13) gives the JAW interval (8) by setting $\widehat{S}(x, y) = y - \widehat{\mu}(x)$, $V_i^L = \widehat{\mu}_{-i}(X_{n+1}) - \widehat{\mu}(X_{n+1}) - R_i^{LOO}$, and $V_i^U = \widehat{\mu}_{-i}(X_{n+1}) - \widehat{\mu}(X_{n+1}) + R_i^{LOO}$; see Appendix B.3. Similarly, (13) gives the prediction interval for weighted split conformal prediction [Tibshirani et al., 2019] for absolute value residual scores when $\widehat{S}(x, y) = |y - \widehat{\mu}(x)|$, and for all calibration data $i$ we let $V_i^U = |Y_i - \widehat{\mu}(X_i)|$ and $V_i^L = 0$.) Then, defining

$$\alpha_E^{\text{w-audit}} = \min \left( \left\{ \alpha' : \tau^- \le \widehat{Q}_{\alpha'}^- \{p_i^w(X_{n+1})\delta_{V_i^L}\}, \ \widehat{Q}_{1-\alpha'}^+ \{p_i^w(X_{n+1})\delta_{V_i^U}\} \le \tau^+ \right\} \right), \tag{14}$$

we can estimate the probability of $\widehat{\mu}(X_{n+1})$ *not* resulting in an error as in (12) as:

$$\widehat{p}\{Y_{n+1} \in \overline{E}\} = \begin{cases} 1 - \alpha_E^{\text{w-audit}} & \text{if } \alpha_E^{\text{w-audit}} \text{ exists} \\ 0 & \text{otherwise.} \end{cases} \tag{15}$$

While the target coverage for $\widehat{C}_{n,\alpha_E}^{\text{w-audit}}(X_{n+1})$ is used in (15), the following theorem gives the worst-case error assessment guarantee for covariate shift (proof in Appendix C.3). Corollary 1 in Appendix B.3 and Corollary 2 in Appendix B.4 give the error assessment guarantees for JAW-E and JAWA-E respectively. Appendix B.2 gives the analogous guarantee for exchangeable data.

**Theorem 3.** *Assume a predictive inference method of the form* (13) *has coverage guarantee* $\mathbb{P}\{Y_{n+1} \in \widehat{C}_{n,\alpha}^{\text{w-audit}}(X_{n+1})\} \ge 1 - c_1\alpha - c_2$ , *with* $c_1, c_2 \in \mathbb{R}$, *under covariate shift* (2) *where* $\widetilde{P}_X$ *is absolutely continuous with respect to* $P_X$. *Define* $\overline{E}$ *as in* (12) *and* $\alpha_E^{\text{w-audit}}$ *as in* (14). *Then,*

$$\mathbb{P}\{Y_{n+1} \in \overline{E}\} \ge \begin{cases} 1 - c_1\alpha_E^{\text{w-audit}} - c_2 & \text{if } \alpha_E^{\text{w-audit}} \text{ exists and } \alpha_E^{\text{w-audit}} < \frac{1-c_2}{c_1} \\ 0 & \text{otherwise.} \end{cases} \tag{16}$$

# 4 Experiments[1]

## 4.1 Datasets and creation of covariate shift

We conduct experiments on five UCI datasets Dua and Graff [2017] with various dimensionality (Table 2): airfoil self-noise, red wine quality prediction [Cortez et al., 2009], wave energy converters, superconductivity [Hamidieh, 2018], and communities and crime [Redmond and Baveja, 2002].

Table 2: Statistics for the UCI datasets. Only the first 2000 samples were used for the wave and superconductivity datasets (for wave, the first 2000 samples of Adelaide data).

| Dataset | # of samples | # of features | Label range |
|---|---|---|---|
| Airfoil self-noise (airfoil) | 1503 | 5 | [103.38, 140.987] |
| Red wine quality (wine) | 1599 | 11 | [3, 8] |
| Wave energy converters (wave) | 2000 | 48 | [1226969, 1449349] |
| Superconductivity (superconduct) | 2000 | 81 | [0.2, 136.0] |
| Communities and crime (communities) | 1994 | 99 | [0, 1] |

We use exponential tilting to induce covariate shift on the test data, based on the approach used in Tibshirani et al. [2019]. We first randomly sample 200 points for the training data, and then sample the biased test data from the remaining datapoints that are not used for training with probabilities proportional to exponential tilting weights. See Appendix D.1 for additional details.

## 4.2 Baselines

**Baselines for comparison to JAW** We compared JAW to the following baselines:

1. **Naive** estimates are based on training data residuals $|Y_i - \widehat{\mu}(X_i)|$, which suffers from overfitting.
2. **Jackknife** uses the classic Jackknife resampling as in (4).
3. **Jackknife+** follows (5), which replaces the prediction $\widehat{\mu}(X_{n+1})$ in jackknife with $\widehat{\mu}_{-i}(X_{n+1})$.
4. **Jackknife-mm** [Barber et al., 2021] is a more conservative alternative to jackknife+ that guarantees coverage at the $1 - \alpha$ level with exchangeable data, but usually with overly-wide intervals.

$$\widehat{C}_{n,\alpha}^{\text{jackknife-mm}}(X_{n+1}) = \left[ \min_{i=1,\ldots,n} \widehat{\mu}_{-i}(X_{n+1}) - Q_{1-\alpha}^+\{R_i^{LOO}\}, \max_{i=1,\ldots,n} \widehat{\mu}_{-i}(X_{n+1}) + Q_{1-\alpha}^+\{R_i^{LOO}\} \right]$$

5. **Cross validation+** (CV+) [Barber et al., 2021] is similar to jackknife+ but splits data into $K$ folds and replaces the $\widehat{\mu}_{-i}(X_{n+1})$ with $\widehat{\mu}_{-k}(X_{n+1})$, the model trained with the $k$th subset removed.
6. **Split** method follows split conformal prediction, which uses half the data for training and the other half for generating the nonconformity scores.
7. **Weighted split** is a version of split conformal with likelihood ratio weights to maintain coverage under covariate shift, as in Tibshirani et al. [2019].

**Baselines for comparison to JAWA** For influence function orders $K \in \{1, 2, 3\}$, we compared the proposed JAWA-$K$ method with $K$-th order influence function approximations of the jackknife-based baselines that we used as comparisons to JAW—we thus refer to these approximations as IF-$K$ jackknife, IF-$K$ jackknife+, and IF-$K$ jackknife-mm. Each baseline compared to JAWA-$K$ is thus also approximated with the same $K$-th order leave-one-out influence function models.

## 4.3 Experimental results

We report experimental results on the predictive interval-generation task for both JAW and JAWA and on the error assessment task for JAW, compared to baselines. Additional experimental details and supplementary experiments can be found in Appendix D, including for estimated likelihood ratio weights in D.5, ablation study with shift magnitudes in D.6, and coverage histograms in D.9.

### 4.3.1 Interval generation results for JAW: Coverage and interval width

Figure 3 compares JAW and its baselines, firstly regarding mean coverage and secondly regarding median interval width, on all five UCI datasets for both neural network and random forest predictors,

---

[1]Additional analysis in Appendix D and code at `https://github.com/drewprinster/jaws.git`.

averaged over 1000 experimental replicates. See Appendix D.2 for predictor function details. Meeting the target coverage level of $1 - \alpha$ is the primary goal of the interval-generation audit task, but for methods that meet or nearly meet the target coverage level, smaller interval widths are more informative. Additionally, smaller variance in coverage indicates a more reliable or consistent method.

As seen in Figure 3, the JAW predictive interval coverage is above the target level of 0.9 across all datasets, for both random forest and neural network $\widehat{\mu}$ functions, along with the jackknife-mm and weighted split methods. However, JAW's interval widths are generally smaller and thus more informative than those of jackknife-mm (which are often overly large, as noted in Barber et al. [2021]). Weighted split and JAW perform similarly on mean coverage and median interval width (both methods have coverage guarantees under covariate shift), but JAW avoids sample splitting and as a result has lower coverage variance than weighted split for all dataset and predictor conditions (see Appendix D.3), which suggests that JAW's predictive intervals are more reliable.

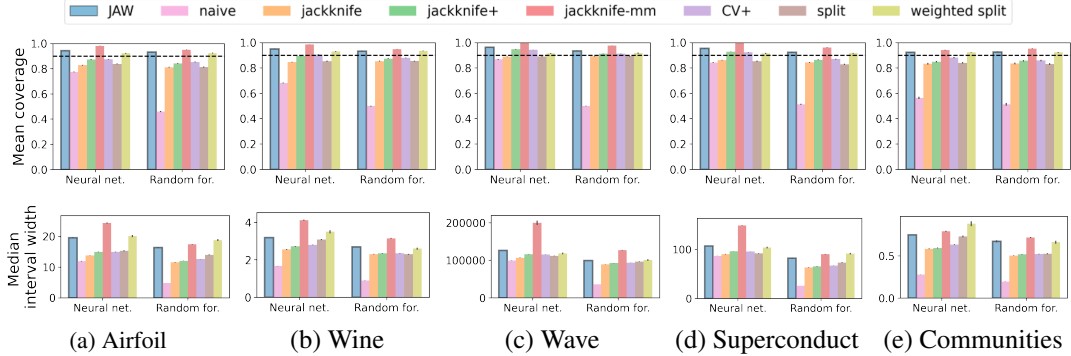

Figure 3: Mean coverage (first row) and median interval width (second row) for neural network and random forest predictors on UCI datasets. Dashed line is the target coverage level ($1 - \alpha = 0.9$). Error bars show the standard error of 1000 repeated experiments. JAW maintains target coverage under covariate shift for all predictor and dataset conditions along with jackknife-mm and weighted split—however, JAW's intervals are generally smaller and thus more informative than jackknife-mm's, and JAW's coverage variance is smaller and thus more reliable than weighted split's (Appendix D.3).

### 4.3.2    Interval generation results for JAWA: Coverage and interval width

Figure 4 evaluates JAWA coverage and interval width compared to baselines for IF orders $K \in \{1, 2, 3\}$ with a neural network predictor (see Appendix D.2 for predictor details). As with the JAW experiments, coverage at the target level of $1 - \alpha = 0.9$ is the primary goal, while secondarily, smaller intervals are more informative for methods that meet or nearly meet target coverage. For three of the five datasets (airfoil, wine, and communities), JAWA is the only method that consistently reaches or nearly reaches the target coverage level. JAWA and all the baselines perform well on the wave datasets, and in the superconduct dataset JAWA still outperforms approximations of jackknife and jackknife+ for all IF orders. Appendix D.7 provides an example empirical comparison of JAWA and JAW runtimes, which demonstrates that JAWA can be orders of magnitude faster to compute.

### 4.3.3    Error assessment results for JAW-E: AUC

We now turn to an error-assessment audit task where the goal is to evaluate a method's ability to estimate the probability that a given prediction is erroneous or not, based on the error criterion $|Y_{n+1} - \widehat{\mu}(X_{n+1})| > \tau$. Let $\overline{E} = [\widehat{\mu}(X_{n+1}) - \tau, \widehat{\mu}(X_{n+1}) + \tau]$. Then, the goal is to estimate the probability that $\widehat{\mu}(X_{n+1})$ is correct, i.e., $Y_{n+1} \in \overline{E}$; or an error, i.e., $Y_{n+1} \notin \overline{E}$. For five predictive interval-generation methods repurposed to the error assessment task (JAW-E, jackknife+E, cross validation+E, split conformal-E, and weighted split conformal-E), Figure 5 reports the area under the receiver operating characteristic curve (AUROC) for 50 repeated experiments with a neural network predictor, with dataset-specific values of $\tau$ (see Appendix D.4 for details and additional experiments with random forest predictor). Better performing methods have higher AUROC values for all values $\tau$. For most tolerance levels and datasets, JAW achieves AUROC values comparable to jackknife+ and CV+ as well as higher AUROC values than split and weighted split conformal prediction. The

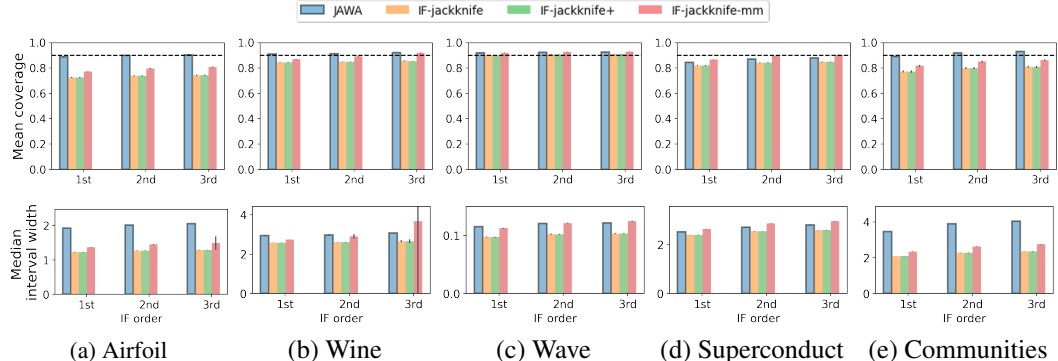

(a) Airfoil     (b) Wine     (c) Wave     (d) Superconduct     (e) Communities

Figure 4: Mean coverage (first row) and median interval width (second row) for JAWA and baselines for influence function orders $K \in \{1, 2, 3\}$. Dashed line is the target coverage level $(1 - \alpha = 0.9)$. Error bar shows the standard error of 200 repeated experiments. JAWA is more consistent than baselines in reaching or nearly reaching the target coverage level across datasets and influence function orders, and it is more computationally efficient than JAW (Appendix D.7).

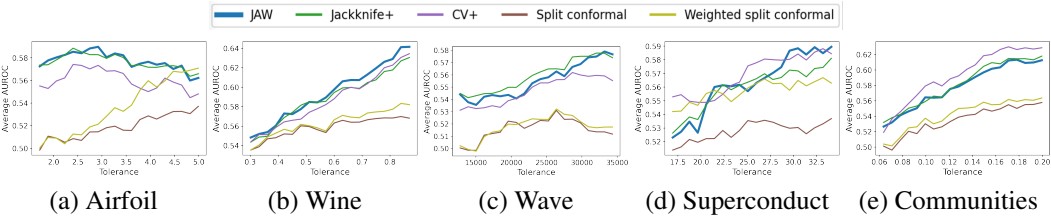

(a) Airfoil     (b) Wine     (c) Wave     (d) Superconduct     (e) Communities

Figure 5: AUROC values for tolerance levels $\tau$ across the three datasets for the neural net predictor, averaged across 50 experiment replicates. Results for random forest predictor in Appendix D.4.

comparable performance of JAW and jackknife+ is likely due to a tradeoff between the benefit of JAW's validity under covariate shift and its reduced effective sample size inherent to likelihood-ratio weighting, as jackknife+'s and CV+'s AUROC degrades with reduced sample size (Appendix D.6).

## 5 Conclusion

In this paper, we develop JAWS, a series of wrapper methods for distribution-free predictive uncertainty auditing tasks when the data exchangeability assumption is violated due to covariate shift. We also propose a general approach to repurposing any distribution-free predictive inference method to the error assessment task. We provide rigorous finite-sample guarantees for JAW and JAW-E on the interval generation and error assessment tasks respectively, and analogous asymptotic guarantees for the computationally efficient JAWA and JAWA-E. We moreover demonstrate superior performance of the JAWS series on a variety of datasets. In supplementary experiments we investigate a number of JAWS' limitations: weight estimation can address the assumed access to oracle weights with similar empirical performance (Appendix D.5), and JAW's increased coverage variance with covariate shift can be explained by reduced effective sample size due to importance weighting (Appendix D.6). Additionally, we note that JAW and JAWA share a limitation with weighted conformal prediction [Tibshirani et al., 2019] of potentially producing overly large intervals in extreme covariate shift cases where a test point's normalized likelihood ratio approaches or exceeds $\alpha$. In the future, we aim to address the problems of reducing coverage variance and improving predictive interval sharpness.

## Acknowledgments

This work was supported by the National Science Foundation grant IIS-1840088. We thank Yoav Wald for helpful discussions and advice, as well as Peter Schulam for sharing code that facilitated our influence function approximation implementation and AUC experiments.

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
