# Supplement to "JAWS: Auditing Predictive Uncertainty Under Covariate Shift"

**Drew Prinster**
Department of Computer Science
Johns Hopkins University
Baltimore, MD 21211
drew@cs.jhu.edu

**Anqi Liu**
Department of Computer Science
Johns Hopkins University
Baltimore, MD 21211
aliu@cs.jhu.edu

**Suchi Saria**
Department of Computer Science
Johns Hopkins University
Baltimore, MD 21211
ssaria@cs.jhu.edu

36th Conference on Neural Information Processing Systems (NeurIPS 2022).

# A  Supplementary background details

## A.1  Error assessment motivation: Concrete example

The error-assessment approach to predictor auditing may be more actionable than the interval-generation approach in safety-critical or high-stakes decision-making situations where there is a clearly defined margin of error that is considered safe or acceptable. One example is chemical or radiation therapy dose prediction for cancer treatment, where administering the correct dosage within $5\% - 10\%$ is safety-critical. Machine learning is being increasingly employed in cancer chemotherapy and radiotherapy for purposes including dose optimization [Feng et al., 2018, Huynh et al., 2020]. Dose errors are one of the most common types of errors in chemotherapy and radiotherapy, occurring when a patient is given a substantially higher or lower than optimal amount of chemical or radiation treatment [Weingart et al., 2018, Van Herk, 2004]. An overdose of either chemotherapeutics or radiation can be harmful or even lethal to a patient, whereas underdose can result in a reduced anticancer effect [Gurney, 2002]. A dose error is generally defined as a percentage deviation, between an administered dose and the truly optimal dose that should have been administered, beyond some error tolerance: $5\%$ or $10\%$ are commonly used deviation thresholds for defining errors [Cohen et al., 1996, Van Herk, 2004]. Accordingly, the probability that the optimal dosage level lies within say $10\%$ of the predicted dosage level may be of greater interest to a provider and their patient than identifying a predictive interval with some predetermined coverage probability (which would provide no error assurance whenever the predictive interval extends beyond the safe threshold, say of $\pm 10\%$).

## A.2  Supplementary background on covariate shift

Covariate shift is a type of dataset shift where the $Y|X$ distribution is the same between training and test data but the marginal $X$ distributions are allowed to change [Sugiyama et al., 2007, Shimodaira, 2000]. This is a strong but common assumption for many dataset shift problems. Covariate shift is also closed related to data missingness and sample selection bias [Bickel et al., 2009]. The most prevalent method for correcting the shift is by applying likelihood ratio or "importance" weights [Sugiyama et al., 2007, Shimodaira, 2000]. Density ratio estimation is then a key subproblem of covariate shift correction [Sugiyama et al., 2012]. Other methods dealing with covariate shift include matching the (kernel) representation between the two distributions [Gretton et al., 2009, Yu and Szepesvári, 2012, Zhang et al., 2013, Zhao et al., 2021] and robust optimization [Liu and Ziebart, 2014, Chen et al., 2016, Duchi et al., 2019, Rezaei et al., 2021].

## A.3  Supplementary comparison to Barber et al. [2022]

In the Section 2.3 of the main paper we contrast our JAW method to the results in Barber et al. [2022] regarding the nonexchangeable jackknife+. We emphasize that Barber et al. [2022] uses fixed weights and compensates for unknown violations of exchangeability at the expense of a coverage gap, whereas JAW uses data-dependent likelihood ratio weights and assumes covariate shift but does not suffer from a coverage gap. Additionally, it is also important to take note of and contrast our work with an extension of the framework in Barber et al. [2022] to data-dependent weights that the authors briefly discuss in their Section 5.3, subsection titled "Fixed versus data-dependent weights" (though this extension is not a primary focus of their work). In short, this extension from Barber et al. [2022] does not generalize to JAW beyond giving a trivial coverage guarantee.

In particular, Barber et al. [2022] do not propose a likelihood-ratio weighting of the jackknife+, but if one were to define the weights in their nonexchangeable jackknife+ as data-dependent, likelihood ratio weights like in our JAW method, then the extension discussed in Section 5.3 of Barber et al. [2022] would in general suffer a coverage gap that could approach 1 under covariate shift. That is, under covariate shift assumptions and with $w_i = w(X_i)$ representing the likelihood ratio for datapoint $i$, the conditional total variation distance between the original ordered data $Z = (Z_1, ..., Z_{n+1})$ and the swapped data $Z^i = (Z_1, ..., Z_{i-1}, Z_{n+1}, Z_{i+1}, ..., Z_n)$ can generally approach 1 for nontrivial covariate shift, i.e., $d_{TV}(Z, Z^i | w_1, ..., w_n, t_1, ..., t_{n+1}) \to 1$. This is because under the covariate shift the training data $\{Z_1, ..., Z_n\}$ and test point $Z_{n+1}$ are not exchangeable (they are weighted exchangeable), meaning that the unweighted data distributions $Z$ and $Z^i$ may have arbitrarily large total variation distance. The result would then be the trivial coverage guarantee (i.e., only guaranteeing coverage probability $\geq 0$).

### A.4 Supplemnetary background on influence functions

In this work we implement the algorithm proposed by Giordano et al. [2019a] to compute higher-order influence functions (IFs), so we refer to Giordano et al. [2019a] for more comprehensive details and theory. However, in this supplementary section we provide additional details on basic IFs theory and our use of IFs for the convenience of the interested reader.

For a weight vector variable $\omega \in \mathbb{R}^n$ and a fixed instance of the variable $\omega = \tilde{\omega}$ representing a specific reweighting of the data, let us denote $\widehat{\mu}_{\tilde{\omega}}$ as the refitted model and $\hat{\theta}(\tilde{\omega})$ as the refitted model parameters that would be obtained by retraining the model with data weights $\tilde{\omega}$. With our notation in this section we maintain some similarity to the notation in Giordano et al. [2019a], but we use the Greek character $\omega$ rather than $w$ to disambiguate the IF data weights $\omega$ from the likelihood-ratio weights $w$ introduced in Section 2.3. For the leave-one-out weight vectors that are of primary interest for approximating the jackknife+ and related methods with influence functions, for ease of notation we say that $\tilde{\omega} = -i$ denotes the all ones vector except with zero in the $i$-th component so that $\widehat{\mu}_{-i}$ still denotes the leave-one-out retrained model, and we denote the corresponding leave-one-out parameters as $\hat{\theta}_{-i} = \hat{\theta}(-i)$.

For any specific weights $\tilde{\omega}$, influence functions assume that $\hat{\theta}(\tilde{\omega})$ is a local minimum of the objective function, and thus that $\hat{\theta}(\tilde{\omega})$ is the solution to the following system of equations, where $G$ is the gradient of the objective function with respect to the model parameters:

$$\hat{\theta}(\tilde{\omega}) := \theta \text{ such that } G(\theta, \tilde{\omega}) := \frac{1}{n}\left(g_0(\theta) + \sum_{i=1}^n \tilde{\omega}_i g_i(\theta)\right) = 0, \tag{17}$$

where $g_i(\theta)$ is the gradient of the objective function for datapoint $i$ and $g_0(\theta)$ is a prior or regularization term. For the predictor $\widehat{\mu} = \widehat{\mu}_{1_n}$ trained on the full, original dataset, we have $\tilde{\omega} = 1_n$ and can thus denote the model parameters for $\widehat{\mu}$ as $\hat{\theta} = \hat{\theta}(1_n)$. For a resampling-based uncertainty quantification method like the jackknife+ (or bootstrap, cross validation, or other jackknife methods), retraining the model for each new reweighting of the training data can sometimes be computationally burdensome or prohibitive. In these cases, we can instead estimate $\hat{\theta}(\omega)$ using influence functions to compute a Taylor series expansion in $\omega$ centered at $1_n$ (or more specifically a Von Mises expansion, see Fernholz [2012]). A first-order influence function—which we will denote as $\delta_\omega^1 \hat{\theta}(1_n)$ for consistency with notation in Giordano et al. [2019a]—refers to the first-order directional derivative of the parameters $\hat{\theta}(\omega)$ with respect to the weights $\omega$:

$$\delta_\omega^1 \hat{\theta}(1_n) = \sum_{i=1}^n \left.\frac{\partial \hat{\theta}(\omega)}{\partial \omega_i}\right|_{\omega=1_n} \Delta\omega_i, \tag{18}$$

where $\Delta\omega = \omega - 1_n$ is the direction of change in weights relative to the original weights $1_n$. The first-order influence function $\delta_\omega^1 \hat{\theta}(1_n)$ thus enables a first-order Taylor series approximatinon of $\hat{\theta}(\omega)$, given by

$$\hat{\theta}^{\text{IF-1}}(\omega) := \hat{\theta}(1_n) + \delta_\omega^1 \hat{\theta}(1_n). \tag{19}$$

Computing the influence function $\delta_\omega^1 \hat{\theta}(1_n)$ requires differentiation through the chain rule because $\hat{\theta}(\omega)$ is only implicitly a function of $\omega$ through estimating equation (17). The first-order Taylor series approximation of $\hat{\theta}(\omega)$ given in (19) can then be rewritten as

$$\hat{\theta}^{\text{IF-1}}(\omega) := \hat{\theta}(1_n) - \hat{H}(\hat{\theta})^{-1} G(\hat{\theta})(w - 1_n). \tag{20}$$

where $\hat{H}(\hat{\theta}) = \hat{H}(\hat{\theta}(1_n), 1_n)$ and $G(\hat{\theta}) = G(\hat{\theta}(1_n), 1_n)$ are the Hessian and the gradient of the objective function.

Similarly, higher-order Taylor series approximations of $\hat{\theta}(\omega)$ can be obtained using higher order influence functions $\delta_\omega^k \hat{\theta}(1_n)$, where the $K$-th order Taylor series is given by

$$\hat{\theta}^{\text{IF-}K}(\omega) := \hat{\theta}(1_n) + \sum_{k=1}^K \frac{1}{k!} \delta_\omega^k \hat{\theta}(1_n). \tag{21}$$

Computing $\hat{\theta}^{\text{IF-K}}(\omega)$ requires several assumptions. See Giordano et al. [2019a] for a formal list, but informally we assume that $\hat{\theta}(1_n)$ is the solution to $G(\hat{\theta}(1_n), 1_n) = 0$, that $G(\theta, 1_n)$ is $K + 1$ times continuously differentiable, that the hessian $H(\hat{\theta})$ is strongly positive definite (meaning that the objective function is strongly convex in the neighborhood of the local solution), and the norm of the derivative $d_\theta^k G(\theta, 1_n)$ has a finite upper bound for $1 \leq k \leq K + 1$. In this work, we implement the recursive procedure based on forward-mode automatic differentiation to achieve memory-efficient computation of higher-order directional derivatives Maclaurin et al. [2015] as described in Giordano et al. [2019a].

While Alaa and Van Der Schaar [2020] propose a higher-order IF approximation of the jackknife+, their method assumes exchangeable (e.g., IID) train and test data and offer experiments with only first and second order IF approximations to the jackknife+. Our proposed JAWA sequence extends the IF approximation of the jackknife+ proposed by Alaa and Van Der Schaar [2020] to the setting of covariate shift, and we demonstrate the benefits of this extension on a variety of datasets and orders of influence function approximation.

# B  Supplementary theoretical results

## B.1  JAW with general weighted exchangeability

In this work, we define the JAW prediction interval (8) using likelihood-ratio weights to address covariate shift as in (2) due to the prevalence and applicability of the covariate shift assumption. However, it is also natural to define a more general version of JAW for other instances of weighted exchangeable data (analogously to how Tibshirani et al. [2019] covers covariate shift as a special case of weighted conformal prediction).

That is, denoting $\{Z_1, ..., Z_{n+1}\} = \{(X_1, Y_1), ..., (X_{n+1}, Y_{n+1})\}$, from Tibshirani et al. [2019] we can define

$$p_i^w(z_1, ..., z_{n+1}) = \frac{\sum_{\sigma:\sigma(n+1)=i} \prod_{j=1}^{n+1} w_j(z_{\sigma(j)})}{\sum_\sigma \prod_{j=1}^{n+1} w_j(z_{\sigma(j)})}, \tag{22}$$

which simplifies to the normalized likelihood ratio weights defined in (3) as a special case when $w_1 = ... = w_n$ and $w_{n+1} = w = \frac{d\widetilde{P}_X}{dP_X}$, as shown in the proof for Corollary 1 in Tibshirani et al. [2019]. A more general version of the JAW prediction interval for general weighted exchangeability with weight functions $w_{1:n+1} = \{w_1, ..., w_{n+1}\}$ can then be defined as

$$\widehat{C}_{n,\alpha,w_{1:n+1}}^{\text{general JAW}}(X_{n+1}) = \Big[ Q_\alpha \big\{ p_i^w(Z_1, ..., Z_{n+1}) \cdot \delta_{\hat{\mu}_{-i}(X_{n+1}) - R_i^{LOO}} + p_{n+1}^w(Z_1, ..., Z_{n+1}) \delta_{-\infty} \big\},$$
$$Q_{1-\alpha} \big\{ p_i^w(Z_1, ..., Z_{n+1}) \cdot \delta_{\hat{\mu}_{-i}(X_{n+1}) + R_i^{LOO}} + p_{n+1}^w(Z_1, ..., Z_{n+1}) \delta_\infty \big\} \Big], \tag{23}$$

where $R_i^{LOO} = \big| \hat{\mu}_{-i}(X_i) - Y_i \big|$, with the $p_i^w(Z_1, ..., Z_{n+1})$ defined as in (22), and where $Q_\beta$ denotes the level $\beta$ quantile function.

The JAW coverage proof technique in Appendix C.1 yields the following result after substituting in the general normalized weights $p_i^w(Z_1, ..., Z_{n+1})$ defined in (22) for the normalized likelihood ratio weights $p_i^w(X_{n+1})$ and replacing mentions of "covariate shift" with "weighted exchangeability":

**Theorem S1.** *Assume that $Z_i = (X_i, Y_i) \in \mathbb{R}^d \times \mathbb{R}, i \in \{1, ..., n + 1\}$ are weighted exchangeable with weight functions $w_1, ..., w_{n+1}$. For $\alpha \in (0, 1)$, the generalized JAW prediction interval in (23) satisfies*

$$\mathbb{P}\big\{ Y_{n+1} \in \widehat{C}_{n,\alpha,w_{1:n+1}}^{\text{general JAW}}(X_{n+1}) \big\} \geq 1 - 2\alpha \tag{S1}$$

## B.2  Error assessment assuming exchangeable data

While in Section 3.3 of the main paper we present a general approach to repurposing predictive interval-generating methods with validity under covariate shift to the error assessment task, here we present the analogous results under the exchangeable data assumption. The results in this

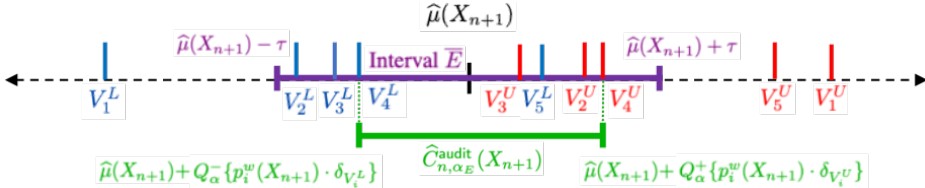

Figure 6: Illustration of terms involved in computing $\alpha_E^{\text{audit}}$ when errors are defined by the event $|Y_{n+1} - \widehat{\mu}(X_{n+1})| > \tau$. The interval $\overline{E} = [\widehat{\mu}(X_{n+1}) - \tau, \widehat{\mu}(X_{n+1}) + \tau]$ is shown in violet, the values $\{V_i^L\}$ in blue, the values $\{V_i^U\}$ in red, and the interval $\widehat{C}_{n,\alpha}^{\text{audit}}(X_{n+1})$ in green. Each vertical line at a location $V_i$ on the real line represents a point mass $\delta_{V_i}$ with height $\frac{1}{n+1}$.

section directly apply to common predictive interval-generating methods including split conformal, jackknife+, and cross-validation+.

The setup is the same as in the main paper Section 3.3, where we first define

$$\overline{E} = \left\{ y \in \mathbb{R} : \tau^- \leq \widehat{S}(X_{n+1}, y) \leq \tau^+ \right\}. \tag{24}$$

However, unlike the covariate shift setting, we instead assume exchangeable data and access to a predictive inference method with valid predictive intervals of the form

$$\widehat{C}_{n,\alpha}^{\text{audit}}(X_{n+1}) = \left\{ y : \widehat{Q}_\alpha^- \left\{ \frac{1}{n+1} \delta_{V_i^L} \right\} \leq \widehat{S}(X_{n+1}, y) \leq \widehat{Q}_{1-\alpha}^+ \left\{ \frac{1}{n+1} \delta_{V_i^U} \right\} \right\} \tag{25}$$

Recall that we use $Q_\alpha^- \{ \frac{1}{n+1} \delta_{V_i^L} \}$ to denote the level $\alpha$ quantile of the empirical distribution $\sum_i^n [ \frac{1}{n+1} \delta_{V_i^L} ] + \delta_{-\infty}$ and $Q_{1-\alpha}^+ \{ \frac{1}{n+1} \delta_{V_i^U} \}$ to denote the level $1 - \alpha$ quantile of the empirical distribution $\sum_i^n [ \frac{1}{n+1} \delta_{V_i^U} ] + \delta_\infty$. (Analogous to as stated in Section 3.3, (25) gives the jackknife+ interval (5) by setting $\widehat{S}(x, y) = y - \widehat{\mu}(x)$, $V_i^L = \widehat{\mu}_{-i}(X_{n+1}) - \widehat{\mu}(X_{n+1}) - R_i^{LOO}$, and $V_i^U = \widehat{\mu}_{-i}(X_{n+1}) - \widehat{\mu}(X_{n+1}) + R_i^{LOO}$. And, (25) gives the prediction interval for split conformal prediction for absolute value residual scores when $\widehat{S}(x, y) = |y - \widehat{\mu}(x)|$, and for all calibration data $i$ we let $V_i^U = |Y_i - \widehat{\mu}(X_i)|$ and $V_i^L = 0$.) Then, define $\alpha_E^{\text{audit}}$ as

$$\alpha_E^{\text{audit}} = \min \left( \left\{ \alpha' : \tau^- \leq \widehat{Q}_{\alpha'}^- \left\{ \frac{1}{n+1} \delta_{V_i^L} \right\}, \widehat{Q}_{1-\alpha'}^+ \left\{ \frac{1}{n+1} \delta_{V_i^U} \right\} \leq \tau^+ \right\} \right). \tag{26}$$

We can then estimate the probability of $\widehat{\mu}(X_{n+1})$ *not* resulting in an error as in (12) as:

$$\widehat{p}\{Y_{n+1} \in \overline{E}\} = \begin{cases} 1 - \alpha_E^{\text{audit}} & \text{if } \alpha_E^{\text{audit}} \text{ exists} \\ 0 & \text{otherwise.} \end{cases} \tag{27}$$

While the target coverage for $\widehat{C}_{n,\alpha_E}^{\text{audit}}(X_{n+1})$ is used in (15), the following theorem gives the worst-case error assessment guarantee for exchangeable data (proof in Appendix C.4).

**Theorem S2.** *If a predictive inference method that generates predictive sets of the form* (25) *has coverage guarantee* $\mathbb{P}\{Y_{n+1} \in \widehat{C}_{n,\alpha}^{\text{audit}}(X_{n+1})\} \geq 1 - c_1 \alpha - c_2$ *assuming exchangeable data, where* $c_1, c_2 \in \mathbb{R}$, *define* $\overline{E}$ *as in* (24) *and* $\alpha_E^{\text{w-audit}}$ *as in* (26). *Then,*

$$\mathbb{P}\{Y_{n+1} \in \overline{E}\} \geq \begin{cases} 1 - c_1 \alpha_E^{audit} - c_2 & \text{if } \alpha_E^{audit} \text{ exists and } \alpha_E^{audit} < \frac{1-c_2}{c_1} \\ 0 & \text{otherwise} \end{cases}. \tag{28}$$

### B.3   JAW-E error assessment guarantee

We now state the error assessment guarantee for JAW-E as Corollary 1, which follows directly from Theorem 3. First, recall that we assume a predictive inference that has valid coverage under covariate shift and can be written in the form of (13), which we restate here:

$$\widehat{C}_{n,\alpha}^{\text{w-audit}}(X_{n+1}) = \left\{ y \in \mathbb{R} : \widehat{Q}_\alpha^- \{ p_i^w(X_{n+1}) \delta_{V_i^L} \} \leq \widehat{S}(X_{n+1}, y) \leq \widehat{Q}_{1-\alpha}^+ \{ p_i^w(X_{n+1}) \delta_{V_i^U} \} \right\} \tag{29}$$

To obtain the JAW predictive interval from (29), we define the test point score function[2] as $\widehat{S}(x,y) = y - \widehat{\mu}(x)$, and for all $i \in \{1,...,n\}$ we let $V_i^L = \widehat{\mu}_{-i}(X_{n+1}) - \widehat{\mu}(X_{n+1}) - R_i^{LOO}$ and $V_i^U = \widehat{\mu}_{-i}(X_{n+1}) - \widehat{\mu}(X_{n+1}) + R_i^{LOO}$:

$$\widehat{C}_{n,\alpha}^{\text{w-audit}}(X_{n+1}) = \left\{ y \in \mathbb{R} : \widehat{Q}_\alpha^- \{p_i^w(X_{n+1})\delta_{\widehat{\mu}_{-i}(X_{n+1})-\widehat{\mu}(X_{n+1})-R_i^{LOO}}\} \leq y - \widehat{\mu}(X_{n+1}) \right.$$
$$\leq \widehat{Q}_{1-\alpha}^+ \{p_i^w(X_{n+1})\delta_{\widehat{\mu}_{-i}(X_{n+1})-\widehat{\mu}(X_{n+1})+R_i^{LOO}}\} \Big\}$$
$$= \left\{ y \in \mathbb{R} : \widehat{Q}_\alpha^- \{p_i^w(X_{n+1})\delta_{\widehat{\mu}_{-i}(X_{n+1})-R_i^{LOO}}\} \leq y \right.$$
$$\leq \widehat{Q}_{1-\alpha}^+ \{p_i^w(X_{n+1})\delta_{\widehat{\mu}_{-i}(X_{n+1})+R_i^{LOO}}\} \Big\}$$
$$= \widehat{C}_{n,\alpha}^{\text{JAW}}(X_{n+1}) \tag{30}$$

Then, let us define $\alpha_E^{JAW}$ as:

$$\alpha_E^{JAW} = \min\left( \left\{ \alpha' : \tau^- \leq Q_{\alpha'}^- \{p_i^w(X_{n+1})\delta_{\widehat{\mu}_{-i}(X_{n+1})-R_i^{LOO}}\}, \right. \right.$$

$$\left. \left. Q_{1-\alpha'}^+ \{p_i^w(X_{n+1})\delta_{\widehat{\mu}_{-i}(X_{n+1})+R_i^{LOO}}\} \leq \tau^+ \right\} \right). \tag{31}$$

**Corollary 1.** *Assume data under covariate shift from (2) where $\tilde{P}_X$ is absolutely continuous with respect to $P_X$. Define $\overline{E}$ as in (12) and $\alpha_E^{JAW}$ as in (31). Then,*

$$\mathbb{P}\{Y_{n+1} \in \overline{E}\} \geq \begin{cases} 1 - 2\alpha_E^{JAW} & \text{if } \alpha_E^{JAW} \text{ exists and } \alpha_E^{JAW} < \frac{1}{2} \\ 0 & \text{otherwise} \end{cases} \tag{32}$$

### B.4 JAWA-E error assessment guarantee

Lastly for our theoretical results, we state the error assessment guarantee for JAWA-E as Corollary 2. Whereas Corollary 1 holds for finite samples, Corollary 2 holds in the limit either of the number of samples or in the order of the influence function approximation.

First, define

$$\alpha_E^{\text{JAWA-}K} = \min\left( \left\{ \alpha' : \tau^- \leq Q_{\alpha'}^- \{p_i^w(X_{n+1})\delta_{\widehat{\mu}_{-i}^{\text{IF-}K}(X_{n+1})-R_i^{\text{IF-}K,LOO}}\}, \right. \right.$$

$$\left. \left. Q_{1-\alpha'}^+ \{p_i^w(X_{n+1})\delta_{\widehat{\mu}_{-i}^{\text{IF-}K}(X_{n+1})+R_i^{\text{IF-}K,LOO}}\} \leq \tau^+ \right\} \right). \tag{33}$$

**Corollary 2.** *Let Assumptions 1 - 4 and either Condition 2 or Condition 4 from Giordano et al. [2019a] hold uniformly for all $n$. Assume data under covariate shift from (2) where $\tilde{P}_X$ is absolutely continuous with respect to $P_X$. Define $\overline{E}$ as in (12) and $\alpha_E^{JAWA-K}$ as in (33). Then,*

*Then, as either $n \to \infty$ or as $K \to \infty$, we have*

$$\mathbb{P}\{Y_{n+1} \in \overline{E}\} \geq \begin{cases} 1 - 2\alpha_E^{\text{JAWA-}K} & \text{if } \alpha_E^{\text{JAWA-}K} \text{ exists and } \alpha_E^{\text{JAWA-}K} < \frac{1}{2} \\ 0 & \text{otherwise} \end{cases} \tag{34}$$

---

[2]Note that the test point score function $\widehat{S}(x,y) = y - \widehat{\mu}(x)$ that we use to obtain an alternative definition of the JAW interval in (30) (and could analogously be used to define the jackknife+) has nuanced differences from the score functions used in weighted standard conformal prediction methods (as well as in their unweighted variants). As mentioned in the main paper, (29) yields the weighted split conformal prediction interval for absolute value residual scores when $\widehat{S}(x,y) = |y - \widehat{\mu}(x)|$, and for all holdout calibration data $i$ we let $V_i^U = |Y_i - \widehat{\mu}(X_i)|$ and $V_i^L = 0$—so, we observe that for weighted split conformal prediction $V_i^U = \widehat{S}(X_i, Y_i)$ for all calibration data $i$, and thus $\widehat{S}$ can be understood as a "nonconformity score" as in standard conformal prediction. However, for JAW (and the jackknife+) there is a less clear correspondence between $\widehat{S}(x,y) = y - \widehat{\mu}(x)$ and $\{V_i^U\}$ (or $\{V_i^L\}$). We thus choose to define $\widehat{S}$ as a *test point* score function in an effort to simultaneously maintain greater clarity on its meaning from a user's perspective, maintain intuitive connections to standard conformal prediction methods, and also avoid suggesting that $\{V_i^U\}$ and $\{V_i^L\}$ are directly defined from $\widehat{S}$ in the case of JAW and the jackknife+. It is also worth noting that there may be some score functions for which the jackknife+ and JAW are not defined, in which case the corresponding error assessment methods would not be defined.

# C Proofs for theoretical results

## C.1 Proof of Theorem 1

*Proof.* We use (a) - (d) to denote four setup steps, and we use 1-3 to denote the main steps in the proof. Our first two initial setup steps (a) and (b) are identical to the corresponding setup steps in the proof for Theorem 1 in Barber et al. [2021]:

(a) First, we suppose the hypothetical case where in addition to the training data $\{(X_1, Y_1), ..., (X_n, Y_n)\}$, we also have access to the test point $(X_{n+1}, Y_{n+1})$. For each pair of indices $i, j \in \{1, ..., n+1\}$ with $i \neq j$, we define $\tilde{\mu}_{-(i,j)}$ as the regression function fitted on the training and test data except with the points $i$ and $j$ removed. (We follow the notation in Barber et al. [2021] where $\tilde{\mu}$ rather than $\hat{\mu}$ reminds us that the former is fit on a subset of data $1, ..., n+1$ that may contain the test point $n+1$.) We note that $\tilde{\mu}_{-(i,j)} = \tilde{\mu}_{-(j,i)}$ for any $i \neq j$, and $\tilde{\mu}_{-(i,n+1)} = \hat{\mu}_{-i}$ for any $i = 1, ..., n$.

(b) We also define the same matrix of residuals in Barber et al. [2021], $R \in \mathbb{R}^{(n+1) \times (n+1)}$, with entries

$$R_{ij} = \begin{cases} +\infty & i = j, \\ |Y_i - \tilde{\mu}_{-(i,j)}(X_i)| & i \neq j \end{cases}$$

such that the off-diagonal entries $R_{ij}$ represent the residual for the $i$th datapoint where both $i$ and $j$ are not seen by the regression fitting.

At this point we begin to introduce some changes to the proof in Barber et al. [2021]:

(c) We define a weighted comparison matrix that we call $A^w \in \mathbb{R}^{(n+1) \times (n+1)}$. First, define $A$ as the unweighted comparison matrix in Barber et al. [2021] with entries $A_{ij} = \mathbb{1}\{R_{ij} > R_{ji}\}$ (indicators for the event that, when $i$ and $j$ are excluded from the regression fitting, $i$ has larger residual than $j$), and define $W$ as the diagonal matrix with $W_{ii} = p_i^w(X_{n+1})$. Then, define $A^w = WAW$, so that $A^w$ has entries $A_{ij}^w = p_i^w(X_{n+1}) \cdot p_j^w(X_{n+1}) \cdot \mathbb{1}\{R_{ij} > R_{ji}\}$. For any $i, j \in \{1, ..., n+1\}$, note that $A_{ij}^w > 0$ implies $A_{ji}^w = 0$ for any $i, j \in \{1, ..., n+1\}$. (Moreover, note that in the absence of covariate shift, $p_i^w(X_{n+1}) = p_j^w(X_{n+1}) = \frac{1}{n+1}$ for all $i, j \in \{1, ..., n+1\}$ and the weighted comparison matrix $A^w$ becomes equivalent up to a normalization constant to the unweighted comparison matrix $A$ described in Barber et al. [2021], i.e., with exchangeable data $A^w = A/(n+1)^2$.)

(d) Next, as in Barber et al. [2021] we are interested in identifying points that have unusually large residuals and are thus hard to predict. Barber et al. [2021] defined such points with unusually large residuals as points $i$ where $\mathbb{1}\{R_{ij} > R_{ji}\}$ for a sufficiently large fraction of other points $j$. However, in the covariate shift setting, we need to account for the fact that the informativeness of the comparison $\mathbb{1}\{R_{ij} > R_{ji}\}$ depends on the likelihood of $j$ in the test distribution relative to the training distribution: If $w(X_j) > w(X_{j'})$ for some points $j, j' \in \{1, ..., n+1\}\setminus i$, $j \neq j'$, then the comparison $\mathbb{1}\{R_{ij} > R_{ji}\}$ should contain more information about how difficult $i$ is to predict than the comparison $\mathbb{1}\{R_{ij'} > R_{j'i}\}$. In particular, we are interested in identifying points $i$ where $\mathbb{1}\{R_{ij} > R_{ji}\}$ for a sufficiently large *total normalized weight* of other points $j$. With this motivation, we here define the set of "strange" points $\mathcal{S}(A^w) \subseteq \{1, ..., n+1\}$ in the following two equivalent ways that each serve a different illustrative purpose:

$$\mathcal{S}(A^w) = \left\{ i \in \{1, ..., n+1\} \; : \; w(X_i) > 0, \quad \sum_{j=1}^{n+1} \left( p_j^w(X_{n+1}) \cdot \mathbb{1}\{R_{ij} > R_{ji}\} \right) \geq 1 - \alpha \right\}$$

$$= \left\{ i \in \{1, ..., n+1\} \; : \; w(X_i) > 0, \quad \frac{\sum_{j=1}^{n+1} A_{ij}^w}{p_i^w(X_{n+1})} \geq 1 - \alpha \right\}$$

The first definition represents our intuition of $\mathcal{S}(A^w)$ as a set of "strange" points, which we have described (where $\mathbb{1}\{R_{ij} > R_{ji}\}$ for a sufficiently large total normalized weight of other points $j$). That is, in the first definition it is relatively straightforward to see

how $\mathcal{S}(A^w) \subseteq \{1, ..., n + 1\}$ is the set of points $i \in \{1, ..., n + 1\}$ such that for all the points $j \in \{1, ..., n + 1\}, j \neq i$ where $R_{ij} > R_{ji}$, that the sum of the normalized weights $p_j^w(X_{n+1})$ of all such points $j$ is sufficiently large (at least $1 - \alpha$). On the other hand, the second definition represents how the set of strange points can be computed from the weighted comparison matrix $A^w$, where $\sum_{j=1}^{n+1} A_{ij}^w / p_i^w(X_{n+1})$ is the row sum of the $i$th row in $A^w$ divided by the common factor of the $i$th row, $p_i^w(X_{n+1})$. (In the absence of covariate shift when $w(X_k) = 1$ for all $k \in \{1, ..., n + 1\}$, these definitions are equivalent to the set of strange points in the jackknife+ coverage proof in Barber et al. [2021].)

The following main steps in our proof take the following structure similar to as in Barber et al. [2021], but generalizing each step to covariate shift:

- Step 1: Establish deterministically that $\sum_{i \in \mathcal{S}(A^w)} p_i^w(X_{n+1}) \leq 2\alpha$. That is, for any comparison matrix $A^w$, it is impossible to have the total normalized weight of all the strange points exceed $2\alpha$.

- Step 2: Using the fact that the datapoints are weighted exchangeable, show that the probability that the test point $n + 1$ is strange (i.e., $n + 1 \in \mathcal{S}(A^w)$) is thus bounded by $2\alpha$.

- Step 3: Lastly, verify that the JAW interval can only fail to cover the test label value $Y_{n+1}$ if $n + 1$ is a strange point.

*Step 1: Bounding the total normalized weight of the strange points.* This proof step follows and generalizes the corresponding proof step for Theorem 1 in Barber et al. [2021], which relies on Landau's theorem for tournaments [Landau, 1953]. For each pair of points $i$ and $j$ where $i \neq j$, let us say that $i$ "wins" its game against point $j$ if $A_{ij}^w > 0$, that is if both $i$ and $j$ have nonzero density in the test distribution and if there is a higher residual on point $i$ than on point $j$ for the regression model $\tilde{\mu}_{-(i,j)}$. We say that $i$ loses its game with $j$ otherwise.

However, whereas Barber et al. [2021] derive a bound on the *number* of strange points from a bound on the *number of pairs* of strange points, we instead derive a bound on the *total normalized weight* of the strange points from a bound on the sum of the *product of normalized weights* for two strange points in a pair. As we will see, this idea generalizes the idea of counting pairs of points to account for continuous weights on the points: If all points have uniform unnormalized weight of 1, then, after adjusting for a normalizing constant in our construction, the product of unnormalized weights of points in a pair is 1 for all pairs and our construction reduces to bounding the number of distinct pairs of strange points.

Observe that, by the definition of a strange point, the points that each strange point $i \in \mathcal{S}(A^w)$ wins against must have total normalized weight greater than or equal to $(1 - \alpha)$, and thus the points that each strange point $i \in \mathcal{S}(A^w)$ loses to can only have total normalized weight at most $\alpha - p_i^w(X_{n+1})$ (our definition does not allow $i$ to lose to itself). That is:

$$\begin{array}{c} \text{Total normalized weight} \\ \text{of points that } i \text{ loses to} \end{array} = \sum_{j=1}^{n+1} \left( p_j^w(X_{n+1}) \cdot \mathbb{1}\{R_{ij} \leq R_{ji}\} \right) \leq \alpha - p_i^w(X_{n+1})$$

This inequality will help us obtain an upper bound on the sum of the product of normalized weights between strange points in a pair. To aid with intuition, it may be helpful to think about a correspondance between a product of two weights and the area of a rectangle with side lengths equal to each weight value. Suppose that for each strange point $i \in \mathcal{S}(A^w)$ we construct a rectangle $L_i$ with width equal point $i$'s normalized weight, $p_i^w(X_{n+1})$, and length equal to the largest total normalized weight that the points that $i$ loses to could have, $\alpha - p_i^w(X_{n+1})$. In addition, suppose that we also construct a second rectangle $L_i'$ for each strange point $i \in \mathcal{S}(A^w)$ with width equal to $p_i^w(X_{n+1})$—note that $L_i'$ has the same width as $L_i$—but with length equal to half the total normalized weight of all of the strange points other than $i$, that is, $\frac{1}{2} \sum_{j \in \mathcal{S}(A^w) \setminus i} p_j^w(X_{n+1})$.

We now take a moment to describe the meaning of the total area of the set of rectangles $\{L_i\}$ in a way that we will soon make use of: The total area of $\{L_i\}$ is an upper bound on the sum of products

of normalized weights for all points in a pair where one point is a strange point and the other point is a point that the strange point loses to. To see this, note that by construction the area of any rectangle $L_i$ is the product of point $i$'s normalized weight (i.e., $p_i^w(X_{n+1})$) with an upper bound on the total normalized weight that the points $i$ loses to could have (i.e., $\alpha - p_i^w(X_{n+1})$). Thus, the area of $L_i$ is by construction an upper bound on the product of point $i$'s normalized weight (i.e., $p_i^w(X_{n+1})$) with the total normalized weight of the points that $i$ *actually* loses to. To state with more precise notation that we will use again later, for each point $j$ that $i$ *actually* loses to, let us construct a rectangle $L_{ij}$ with width $p_i^w(X_{n+1})$ and length $p_j^w(X_{n+1})$. Then, for all these points $j$, we can arrange the rectangles $\{L_{ij}\}$ so that they are contained within $L_i$ and so that $L_{ij}$ and $L'_{ij}$ have zero overlapping area for all $j \neq j'$: that is, by this construction $\sum_{j:i \text{ loses to } j} \text{Area}(L_{ij}) \leq \text{Area}(L_i)$. So, it is equivalent to describe the area of $L_i$ as an upper bound on the sum, over all points $j$ that $i$ loses to, of the product of $i$'s normalized weight with $j$'s normalized weight; and thus by extension, the total area of $\{L_i\}$ is as we described earlier.

On the other hand, the total area of the set of rectangles $\{L'_i\}$ is the sum of the product of the normalized weights of two strange points in a pair over all pairs of strange points, where the factor of $\frac{1}{2}$ avoids double counting the pairs of strange points. To see this, note that for every pair of strange points $\{i, j\}$ there is a distinct subrectangle—call it $L'_{ij}$—that is contained in $L'_i$, such that $L'_{ij}$ has width $p_i^w(X_{n+1})$ and length $\frac{1}{2}p_j^w(X_{n+1})$ (where we also assume that for any $j \neq j'$, $L_{ij}$ and $L'_{ij}$ overlapping area of zero). Moreover, for this pair of strange points $\{i, j\}$ there is also an analogous subrectangle $L'_{ji}$ with width $p_j^w(X_{n+1})$ and length $\frac{1}{2}p_i^w(X_{n+1})$ contained in $L'_j$. Thus, the combined area of $L'_{ji}$ and $L'_{ij}$ is $\text{Area}(L'_{ij}) + \text{Area}(L'_{ji}) = p_i^w(X_{n+1}) \cdot p_j^w(X_{n+1})$, and the total area of the set of rectangles $\{L'_i\}$ is as described. (Furthermore, note that when the unnormalized weights are all equal to 1 as in Barber et al. [2021], the area of $\{L'_i\}$—adjusted by a normalization constant—is equivalent to the total number of pairs of strange points $s(s-1)/2$, where $s = |S(A^w)|$ is the number of strange points.)

Now, observe that any pair of two strange points is also a pair of points where one point is strange and the other is a point that the strange point loses to, so the set of pairs of points included in the construction of $\{L'_i\}$ is a subset of the set of pairs of points for which the area of $\{L_i\}$ is the upper bound previously described. To be more precise, let $\{i, j\}$ be a pair of strange points, where (without loss of generality) let us say $i$ loses to $j$. Then, for the $L'_{ij}$ and $L'_{ji}$ as described before, there exists a distinct $L_{ij}$ such that $\text{Area}(L'_{ij}) + \text{Area}(L'_{ji}) = \text{Area}(L_{ij})$. More generally, we see that the total area of all the subrectangles $\{L'_{ij}\}$ is bounded by the total area of the subrectangles $\{L_{ij}\}$, that is $\sum_{i,j \in S(A^w), i \neq j} \text{Area}(L'_{ij}) = \sum_{i,j \in S(A^w), i \neq j} \text{Area}(L_{ij}) \leq \sum_{i \in S(A^w), i \text{ loses to } j} \text{Area}(L_{ij})$. Moreover, by construction $\sum_{i,j \in S(A^w), i \neq j} \text{Area}(L'_{ij}) = \sum_{i \in S(A^w)} \text{Area}(L'_i)$ and $\sum_{i \in S(A^w), i \text{ loses to } j} \text{Area}(L_{ij}) \leq \sum_{i \in S(A^w)} \text{Area}(L_i)$. Therefore, the area of the set of rectangles $\{L'_i\}$ is less than or equal to the area of rectangles $\{L_i\}$, which we can write as follows:

$$\sum_{i \in S(A^w)} \text{Area}(L'_i) \leq \sum_{i \in S(A^w)} \text{Area}(L_i)$$

$$\sum_{i \in S(A^w)} \left( p_i^w(X_{n+1}) \cdot \frac{1}{2} \sum_{j \in S(A^w) \setminus i} p_j^w(X_{n+1}) \right) \leq \sum_{i \in S(A^w)} \left( p_i^w(X_{n+1}) \cdot \left( \alpha - p_i^w(X_{n+1}) \right) \right)$$

$$\text{(C.1.1)}$$

Recall that we defined $p_i^w(X_{n+1}) = w(X_i) / \sum_{k=1}^{n+1} w(X_k) \; \forall \; i \in \{1, ..., n+1\}$, so in the uniform weighted case where $w(X_i) = 1 \; \forall \; i \in \{1, ..., n+1\}$ then $\sum_{k=1}^{n+1} w(X_k) = n+1$, and multiplying both sides of the inequality above by $(n+1)^2$ yields the analogous inequality in Barber et al. [2021] that bounds the number of pairs of points.

We now proceed to solve for an upper bound on $\sum_{i \in \mathcal{S}(A^w)} p_i^w(X_{n+1})$, the total normalized weight of strange points:

$$\frac{1}{2} \sum_{i \in \mathcal{S}(A^w)} \left( p_i^w(X_{n+1}) \cdot \sum_{j \in \mathcal{S}(A^w) \setminus i} p_j^w(X_{n+1}) \right) \leq \sum_{i \in \mathcal{S}(A^w)} \left( p_i^w(X_{n+1}) \cdot \left( \alpha - p_i^w(X_{n+1}) \right) \right)$$

$$\frac{1}{2} \sum_{i,j \in \mathcal{S}(A^w),\ i \neq j} p_i^w(X_{n+1}) p_j^w(X_{n+1}) \leq \alpha \sum_{i \in \mathcal{S}(A^w)} p_i^w(X_{n+1}) - \sum_{i \in \mathcal{S}(A^w)} p_i^w(X_{n+1})^2$$

$$\frac{1}{2} \left( \sum_{i \in \mathcal{S}(A^w)} p_i^w(X_{n+1})^2 + \sum_{i,j \in \mathcal{S}(A^w)} p_i^w(X_{n+1}) p_j^w(X_{n+1}) \right) \leq \alpha \sum_{i \in \mathcal{S}(A^w)} p_i^w(X_{n+1})$$

$$\frac{1}{2} \left( \sum_{i \in \mathcal{S}(A^w)} p_i^w(X_{n+1})^2 + \sum_{i \in \mathcal{S}(A^w)} \left( p_i^w(X_{n+1}) \cdot \sum_{j \in \mathcal{S}(A^w)} p_j^w(X_{n+1}) \right) \right) \leq \alpha \sum_{i \in \mathcal{S}(A^w)} p_i^w(X_{n+1})$$

$$\frac{1}{2} \left( \sum_{i \in \mathcal{S}(A^w)} p_i^w(X_{n+1})^2 + \left( \sum_{i \in \mathcal{S}(A^w)} p_i^w(X_{n+1}) \right)^2 \right) \leq \alpha \sum_{i \in \mathcal{S}(A^w)} p_i^w(X_{n+1})$$

$$\frac{\sum_{i \in \mathcal{S}(A^w)} p_i^w(X_{n+1})^2}{\sum_{i \in \mathcal{S}(A^w)} p_i^w(X_{n+1})} + \left( \sum_{i \in \mathcal{S}(A^w)} p_i^w(X_{n+1}) \right) \leq 2\alpha$$

$$\sum_{i \in \mathcal{S}(A^w)} p_i^w(X_{n+1}) \leq 2\alpha - \frac{\sum_{i \in \mathcal{S}(A^w)} p_i^w(X_{n+1})^2}{\sum_{i \in \mathcal{S}(A^w)} p_i^w(X_{n+1})}$$

where because $0 \leq p_i^w(X_{n+1}) \leq 1 \ \forall\, i = 1, ..., n+1$ and $p_i^w(X_{n+1}) > 0$ for some $i \in \{1, ..., n+1\}$, we have $0 \leq p_i^w(X_{n+1})^2 \leq p_i^w(X_{n+1}) \ \forall\, i = 1, ..., n+1$ and thus $0 \leq \frac{\sum_{i \in \mathcal{S}(A^w)} p_i^w(X_{n+1})^2}{\sum_{i \in \mathcal{S}(A^w)} p_i^w(X_{n+1})} \leq 1$, and we have

$$\sum_{i \in \mathcal{S}(A^w)} p_i^w(X_{n+1}) \leq 2\alpha \tag{35}$$

as desired.

*Step 2: Weighted exchangeability of the datapoints.* We now leverage the weighted exchangeability of the data to show that, since the total weight of the strange points is at most $2\alpha$, that a test point has at most $2\alpha$ probability of being strange. We organize this step into the following pieces:

- Step 2.1: Argue that $A^w \overset{\mathrm{d}}{=} P_\pi A^w P_\pi^\top$ for any $(n+1) \times (n+1)$ permutation matrix $P_\pi$.

- Step 2.2: Argue that $\mathbb{P}\{n+1 \in \mathcal{S}(A^w)\} = \mathbb{P}\{j \in \mathcal{S}(A^w)\}$ for all $j \in \{1, ..., n+1\}$.

- Step 2.3: Use the fact that the total weight of the strange points is at most $2\alpha$ (from Step 1) to show that $\mathbb{P}\{n+1 \in \mathcal{S}(A^w)\} \leq 2\alpha$.

Beginning with Step 2.1, observe that with $W$ denoting the diagonal matrix with $W_{ii} = p_i^w(X_{n+1})$, $WA$ has entries $(WA)_{ij} = p_i^w(X_{n+1}) \cdot \mathbb{1}\{R_{ij} > R_{ji}\}$ (equivalent to $A$ with each $i$th row weighted by $p_i^w(X_{n+1})$); that $AW$ has entries $(AW)_{ij} = p_j^w(X_{n+1}) \cdot \mathbb{1}\{R_{ij} > R_{ji}\}$ (equivalent to $A$ with each $j$th column weighted by $p_j^w(X_{n+1})$); and recall that $A^w = WAW$. For a permutation $\pi$ of $\{1, ..., n+1\}$, let $P_\pi$ denote the corresponding permutation matrix—that is, $\pi(i') = i \iff P_\pi(i', i) = 1$, which corresponds to the $i$th row in $A$ becoming the $i'$th row in $P_\pi A$. With $\overset{\mathrm{d}}{=}$ denoting equality in distribution, we will argue that $P_\pi WA \overset{\mathrm{d}}{=} WA$ and $AWP_\pi^\top \overset{\mathrm{d}}{=} AW$, which together implies $P_\pi A^w P_\pi^\top \overset{\mathrm{d}}{=} A^w$.

To show $P_\pi WA \overset{\mathrm{d}}{=} WA$, we draw on and adapt ideas from the proof for Lemma 3 in Tibshirani et al. [2019]. For simplicity we assume that the pairs $(R_{ij}, R_{ji})$ are distinct almost surely (the result holds in the general case as well, but the notation is more cumbersome). Using condensed notation for the data as $\{Z_1, ..., Z_{n+1}\} = \{(X_1, Y_1), ..., (X_{n+1}, Y_{n+1})\}$, denote by $E_z$ the event that $\{Z_1, ..., Z_{n+1}\} = \{z_1, ..., z_{n+1}\}$, and let $f$ denote the density function of the joint sample $Z_1, ..., Z_{n+1}$. Note that $P_\pi WA$—which results from permuting the rows of $WA$—does not change

the column membership of any entry in $WA$. In particular, $P_\pi WA$ has entries $(P_\pi WA)_{ij} = (WA)_{\pi(i)j}$, so to show $P_\pi WA \stackrel{d}{=} WA$ it is sufficient to show that each $j$th column in $P_\pi WA$ is equivalent in distribution to the corresponding $j$th column in $WA$. To do so, we begin by conditioning on $E_z$ and then inspecting the probability of the joint event $R_{n+1,j} = r_{ij}, R_{j,n+1} = r_{ji}$ for each $i \in \{1, ..., n+1\}$ in each $j$th column, which occurs when $Z_{n+1} = z_i$:

$$\mathbb{P}\{R_{n+1,j} = r_{ij}, R_{j,n+1} = r_{ji} \mid E_z\} = \mathbb{P}\{Z_{n+1} = z_i \mid E_z\}$$
$$= \frac{\sum_{\pi:\pi(n+1)=i} f(z_{\pi(1)}, ..., z_{\pi(n+1)})}{\sum_\pi f(z_{\pi(1)}, ..., z_{\pi(n+1)})},$$

where the second line above follows by the same reasoning as in the proof for Lemma 3 in Tibshirani et al. [2019]. Then, recalling that data from covariate shift (2) are weighted exchangeable with weight functions $w_1 = ... = w_n = 1$ and $w_{n+1} = w = \frac{d\tilde{P}_X}{dP_X}$, this becomes

$$\mathbb{P}\{R_{n+1,j} = r_{i,j}, R_{j,n+1} = r_{j,i} \mid E_z\} = \frac{\sum_{\pi:\pi(n+1)=i} w(x_{\pi(n+1)}) g(z_{\pi(1)}, ..., z_{\pi(n+1)})}{\sum_\pi w(x_{\pi(n+1)}) g(z_{\pi(1)}, ..., z_{\pi(n+1)})}$$
$$= \frac{\sum_{\pi:\pi(n+1)=i} w(x_{\pi(n+1)}) g(z_1, ..., z_{n+1})}{\sum_\pi w(x_{\pi(n+1)}) g(z_1, ..., z_{n+1})}$$
$$= \frac{\sum_{\pi:\pi(n+1)=i} w(x_{\pi(n+1)})}{\sum_\pi w(x_{\pi(n+1)})}$$
$$= \frac{w(x_i)}{\sum_{k=1}^{n+1} w(x_k)}$$
$$= p_i^w(x_{n+1}),$$

which can be written as

$$(R_{n+1,j}, R_{j,n+1}) \mid E_z \sim \sum_{i=1}^{n+1} p_i^w(x_{n+1}) \delta_{(r_{ij}, r_{ji})}.$$

Due to the conditioning on $E_z$, this is equivalent to

$$(R_{n+1,j}, R_{j,n+1}) \mid E_z \sim \sum_{i=1}^{n+1} p_i^w(X_{n+1}) \delta_{(R_{ij}, R_{ji})},$$

and since this statement holds for any $\{Z_1, ..., Z_{n+1}\} = \{z_1, ..., z_{n+1}\}$, marginalization yields

$$(R_{n+1,j}, R_{j,n+1}) \sim \sum_{i=1}^{n+1} p_i^w(X_{n+1}) \delta_{(R_{ij}, R_{ji})}.$$

More generally, substituting in the index $i'$ for $n+1$ in the argument above yields

$$(R_{i',j}, R_{j,i'}) \sim \sum_{i=1}^{n+1} p_i^w(X_{n+1}) \delta_{(R_{ij}, R_{ji})}. \tag{36}$$

Statement (36) tells us that within each $j$th column, draws of $(R_{i',j}, R_{j,i'})$ from this discrete distribution resemble the analogous draw $(R_{n+1,j}, R_{j,n+1})$ for the test point. That is, the distribution of $(R_{i',j}, R_{j,i'})$ in (36) is irrespective of the index $i'$ and so these draws "look exchangeable", and the distribution of an arbitrary $j$th column of $WA$ does not depend on the ordering of its elements. Thus, $P_\pi WA \stackrel{d}{=} WA$ and by a similar argument $AW P_\pi^\top \stackrel{d}{=} AW$, which together implies $P_\pi A^w P_\pi^\top \stackrel{d}{=} A^w$ for any $(n+1) \times (n+1)$ permutation matrix $P_\pi$, the desired result for Step 2.1.

Because $P_\pi A^w P_\pi^\top \stackrel{d}{=} A^w$ from Step 2.1, this implies $\mathbb{P}\{j \in \mathcal{S}(P_\pi A^w P_\pi^\top)\} = \mathbb{P}\{j \in \mathcal{S}(A^w)\}$. Now, let $P_\pi$ denote a specific permutation matrix that maps $n+1$ to $j$, that is where $P_\pi(j, n+1) = 1$. Then, deterministically, $n+1 \in \mathcal{S}(A^w) \iff j \in \mathcal{S}(\Pi A^w \Pi^\top)$, so we have

$$\mathbb{P}\{n + 1 \in \mathcal{S}(A^w)\} = \mathbb{P}\{j \in \mathcal{S}(P_\pi A^w P_\pi^\top)\} = \mathbb{P}\{j \in \mathcal{S}(A^w)\}$$

for all $j = 1, ..., n + 1$. That is, an arbitrary training point $j$ is equally likely to be strange as the test point $n + 1$, which concludes Step 2.2.

Then, we begin Step 2.3 by multiplying by $p_j^w(X_{n+1})$ to obtain

$$p_j^w(X_{n+1}) \cdot \mathbb{P}\{n + 1 \in \mathcal{S}(A^w)\} = p_j^w(X_{n+1}) \cdot \mathbb{P}\{j \in \mathcal{S}(A^w)\}$$

And summing over $j$, we have

$$\sum_{j=1}^{n+1} p_j^w(X_{n+1}) \cdot \mathbb{P}\{n + 1 \in \mathcal{S}(A^w)\} = \sum_{j=1}^{n+1} p_j^w(X_{n+1}) \cdot \mathbb{P}\{j \in \mathcal{S}(A^w)\}$$

$$\mathbb{P}\{n + 1 \in \mathcal{S}(A^w)\} \cdot \sum_{j=1}^{n+1} p_j^w(X_{n+1}) = \sum_{j=1}^{n+1} p_j^w(X_{n+1}) \cdot \mathbb{P}\{j \in \mathcal{S}(A^w)\}$$

$$\mathbb{P}\{n + 1 \in \mathcal{S}(A^w)\} = \sum_{j=1}^{n+1} p_j^w(X_{n+1}) \cdot \mathbb{P}\{j \in \mathcal{S}(A^w)\}$$

$$= \mathbb{E}\left[ \sum_{j \in \mathcal{S}(A^w)} p_j^w(X_{n+1}) \right]$$

$$\leq 2\alpha$$

where the last line follows from Step 1.

*Step 3: Connection to JAW:* We would now like to connect our strange point result from Step 2 to coverage of the JAW prediction interval. Following the approach of Barber et al. [2021], suppose that $Y_{n+1} \notin \widehat{C}_{n,\alpha}^{\text{JAW}}(X_{n+1})$. Then, either

$$Y_{n+1} > Q_{1-\alpha}^+ \big\{ p_i^w(X_{n+1}) \delta_{\widehat{\mu}_{-i}(X_{n+1}) + R_i^{LOO}} \big\}$$

$$\implies \sum_{i=1}^n p_i^w(X_{n+1}) \cdot \mathbb{1}\big\{ Y_{n+1} > \widehat{\mu}_{-i}(X_{n+1}) + R_i^{LOO} \big\} \geq 1 - \alpha$$

or otherwise

$$Y_{n+1} < Q_\alpha^- \big\{ p_i^w(X_{n+1}) \delta_{\widehat{\mu}_{-i}(X_{n+1}) + R_i^{LOO}} \big\}$$

$$\implies \sum_{i=1}^n p_i^w(X_{n+1}) \cdot \mathbb{1}\big\{ Y_{n+1} < \widehat{\mu}_{-i}(X_{n+1}) - R_i^{LOO} \big\} \geq 1 - \alpha$$

And we can write the union of these two events as

$$1 - \alpha \leq \sum_{i=1}^n p_i^w(X_{n+1}) \cdot \mathbb{1}\big\{ Y_{n+1} \notin \widehat{\mu}_{-i}(X_{n+1}) \pm R_i^{LOO} \big\}$$

$$= \sum_{i=1}^n p_i^w(X_{n+1}) \cdot \mathbb{1}\big\{ \big| Y_i - \widehat{\mu}_{-i}(X_i) \big| < \big| Y_{n+1} - \widehat{\mu}_{-i}(X_{n+1}) \big| \big\}$$

$$= \sum_{i=1}^{n+1} p_i^w(X_{n+1}) \cdot \mathbb{1}\big\{ R_{i,n+1} < R_{n+1,i} \big\}$$

from which we see that $n + 1 \in \mathcal{S}(A^w)$—that is, $n + 1$ is a strange point. This result together with the result from Step 2 gives us

$$\mathbb{P}\big\{ Y_{n+1} \notin \widehat{C}_{n,\alpha}^{\text{JAW}}(X_{n+1}) \big\} \leq \mathbb{P}\big\{ n + 1 \in \mathcal{S}(A^w) \big\} \leq 2\alpha$$

$$\therefore \; \mathbb{P}\big\{ Y_{n+1} \in \widehat{C}_{n,\alpha}^{\text{JAW}}(X_{n+1}) \big\} \geq 1 - 2\alpha$$

$\square$

## C.2  Proof of Theorem 2

*Proof.* First, assume that Assumptions 1 - 4 and Condition 2 from Giordano et al. [2019a] hold uniformly for all $n$ (where $n$ is the number of training points). Then, Proposition 1 from Giordano et al. [2019a] establishes that

$$\max_{i \in [n]} \left\| \hat{\theta}_{-i}^{\text{IF-}K} - \hat{\theta}_{-i} \right\|_2 = O_p(N^{-\frac{1}{2}(K+1)}) \tag{37}$$

So, for fixed $K$:

$$\lim_{N \to \infty} \max_{i \in [n]} \left\| \hat{\theta}_{-i}^{\text{IF-}K} - \hat{\theta}_{-i} \right\|_2 = O_p(N^{-\frac{1}{2}(K+1)}) = 0 \tag{38}$$

Or, for fixed $N$:

$$\lim_{K \to \infty} \max_{i \in [n]} \left\| \hat{\theta}_{-i}^{\text{IF-}K} - \hat{\theta}_{-i} \right\|_2 = O_p(N^{-\frac{1}{2}(K+1)}) = 0 \tag{39}$$

Thus, $\hat{\theta}_{-i}^{\text{IF-}K} \to \hat{\theta}_{-i}$ as either $N \to \infty$ or $K \to \infty$. This implies that $\hat{\mu}_{-i}^{\text{IF-}K} \to \hat{\mu}_{-i}$ as either $N \to \infty$ or $K \to \infty$ because the model $\hat{\mu}_{-i}$ is fully determined by its parameters $\hat{\theta}_{-i}$. Therefore, $\widehat{C}_{n,\alpha}^{\text{JAWA-}K}(X_{n+1}) \to \widehat{C}_{n,\alpha}^{\text{JAW}}(X_{n+1})$ in the limit of $N$ or $K$, and thus by Theorem 1, $\mathbb{P}\{Y_{n+1} \in \widehat{C}_{n,\alpha}^{\text{JAWA-}K}(X_{n+1})\} \geq 1 - 2\alpha$ as $N \to \infty$ or $K \to \infty$.

Now, separately assume that Assumptions 1 - 4 and Condition 4 from Giordano et al. [2019a] hold uniformly for all $n$. Then, Proposition 3 from Giordano et al. [2019a] gives that

$$\max_{i \in [n]} \left\| \hat{\theta}_{-i}^{\text{IF-}K} - \hat{\theta}_{-i} \right\|_2 = O(N^{-(K+1)}) \tag{40}$$

The rest follows from a similar argument as when we assumed Condition 2. $\qquad\square$

## C.3  Proof of Theorem 3

*Proof.* Recall that we define $\overline{E}$ as

$$\overline{E} = \left\{ y \in \mathbb{R} : \tau^- \leq \widehat{S}(X_{n+1}, y) \leq \tau^+ \right\}, \tag{41}$$

that we assume access to a predictive inference method with prediction sets given by

$$\widehat{C}_{n,\alpha}^{\text{w-audit}}(X_{n+1}) = \left\{ y \in \mathbb{R} : \widehat{Q}_\alpha^- \{ p_i^w(X_{n+1})\delta_{V_i^L} \} \leq \widehat{S}(X_{n+1}, y) \leq \widehat{Q}_{1-\alpha}^+ \{ p_i^w(X_{n+1})\delta_{V_i^U} \} \right\} \tag{42}$$

and moreover that we define $\alpha_E^{\text{w-audit}}$ as

$$\alpha_E^{\text{w-audit}} = \min \left( \left\{ \alpha' : \tau^- \leq \widehat{Q}_{\alpha'}^- \{ p_i^w(X_{n+1})\delta_{V_i^L} \}, \ \widehat{Q}_{1-\alpha'}^+ \{ p_i^w(X_{n+1})\delta_{V_i^U} \} \leq \tau^+ \right\} \right). \tag{43}$$

Then, if $\alpha_E^{\text{w-audit}}$ exists and $\alpha_E^{\text{w-audit}} < \frac{1-c_2}{c_1}$, then by construction we can combine (43) with the definition of the prediction set $\widehat{C}_{n,\alpha_E}^{\text{w-audit}}(X_{n+1})$ to obtain

$$\widehat{C}_{n,\alpha_E}^{\text{w-audit}}(X_{n+1}) = \Big\{ y : \ \tau^- \leq \widehat{Q}_{\alpha_E}^- \{ p_i^w(X_{n+1})\delta_{V_i^L} \} \leq \widehat{S}(X_{n+1}, y)$$
$$\leq \widehat{Q}_{1-\alpha_E}^+ \{ p_i^w(X_{n+1})\delta_{V_i^L} \} \leq \tau^+ \Big\}, \tag{44}$$

which shows that $\widehat{C}_{n,\alpha_E}^{\text{w-audit}}(X_{n+1}) \subseteq \overline{E}$. Thus, $\mathbb{P}\{Y_{n+1} \in \overline{E}\} \geq \mathbb{P}\{Y_{n+1} \in \widehat{C}_{n,\alpha_E}^{\text{w-audit}}(X_{n+1})\}$, and by the coverage guarantee for $\widehat{C}_{n,\alpha_E}^{\text{audit}}(X_{n+1})$ it follows that

$$\mathbb{P}\{Y_{n+1} \in \overline{E}\} \geq \mathbb{P}\{Y_{n+1} \in \widehat{C}_{n,\alpha_E}^{\text{w-audit}}(X_{n+1})\} \geq 1 - c_1\alpha_E^{\text{w-audit}} - c_2. \tag{45}$$

Otherwise, $\alpha_E^{\text{w-audit}}$ does not exist or $\alpha_E^{\text{w-audit}} \geq \frac{1-c_2}{c_1} \implies 1 - c_1\alpha_E^{\text{audit}} - c_2 \leq 0$. Neither of these cases yield a nontrivial (positive) lower bound for $\mathbb{P}\{Y_{n+1} \in \overline{E}\}$, so for these cases

$$\mathbb{P}\{Y_{n+1} \in \overline{E}\} \geq \mathbb{P}\{Y_{n+1} \in \widehat{C}_{n,\alpha_E}^{\text{audit}}(X_{n+1})\} \geq 0. \tag{46}$$

$\square$

### C.4 Proof of Theorem B.2

*Proof.* The proof proceeds similarly as the proof for Theorem 3 in Appendix C.3, except replacing the data-dependent weights $p_i^w(X_{n+1})$ with the uniform weights $\frac{1}{n+1}$.

$\square$

## D Additional experimental details and analysis

### D.1 Creation of covariate shift

To induce covariate shift, test points were sampled from the set of points not used for training with exponential tilting weights such that the total number of test points was equal to half the number of points not used for training. For the relatively lower dimensional airfoil and wine datasets, the weights took the form $w(x) = \exp(x^T\beta)$, while for the relatively higher dimensional datasets the weights took the form $w(x) = \exp(x_{\text{PCA}}^T\beta)$ where $x_{\text{PCA}}$ is some PCA-based representation of the covariates data $x$.

Figure 7 shows the distribution first and last features in the airfoil dataset before and after the exponential tilting is applied to induce covariate shift with parameter $\beta = (-1, 0, 0, 0, 1)$. In our main experiments, exponential tilting parameters were selected for each dataset so that the associated covariate shift would result in a similar loss in how informative the training set is regarding the biased test set, as assess by reduced effective sample size.

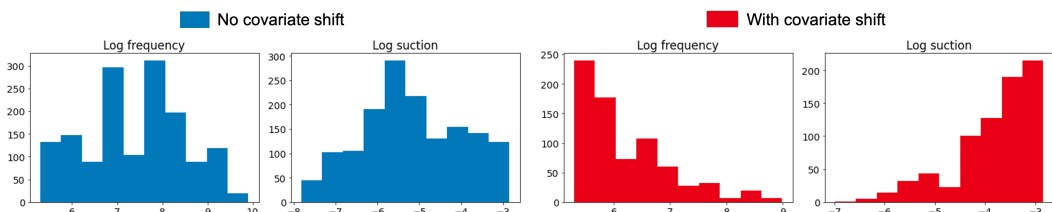

Figure 7: Distribution log frequency and log suction features of airfoil dataset before and after exponential tilting.

Specifically, for a training set size of 200 points for each dataset, the exponential tilting parameters were selected and tuned so that the estimated effective sample size of the training data was reduced to approximately 50, averaged across 1000 random train-test splits. For training data $X_1, ..., X_n$ and likelihood ratio $w$, the effective sample size was estimated using the following commonly-used heuristic $\widehat{n} = [\sum_{i=1}^n |w(X_i)|]^2 / \sum_{i=1}^n |w(X_i)|^2$ [Gretton et al., 2009, Reddi et al., 2015, Tibshirani et al., 2019].

The specific selections of $\beta$ that resulted in approximately $\widehat{n} = 50$ for each dataset are as follows. For the airfoil dataset, unless otherwise specified the tilting parameter was $\beta_{\text{airfoil}} = (-0.85, 0, 0, 0, 0.85)$, which induced covariate shift such that points with low values of the first feature and high values of the last feature were more likely to appear in the test distribution (see Figure 7). The wine dataset was similarly tilted using the first and last components, with a tilting parameter of $\beta_{\text{wine}} =$

$(-0.53, 0, ..., 0, 0.53)$. The wave dataset is composed of 48 total features, of which the first 32 features are latitude and longitude values, and where the remaining 16 features are absorbed power values. Accordingly, the first principal component of only the 32 location features was used for tilting along with the first principal component of only the 16 absorbed power values, with a tilting parameter of $\beta_{\text{wave}} = (-0.0000925, 0.0000925)$ unless otherwise specified. For the superconductivity dataset, only the first principal component of all of the data was used for tilting, with tilting parameter $\beta_{\text{superconduct}} = 0.00062$. Lastly, for the communities and crime dataset, the first two principal components of the whole dataset were used with tilting parameter $\beta_{\text{communities}} = (-0.825, 0.825)$.

## D.2 Models

For all experiments with JAW and its baselines we used two different regression predictors $\widehat{\mu}$: random forests (scikit-learn RandomForestRegressor), and neural networks (scikit-learn MLPRegressor with LBFGS optimizer, logistic activation, and default parameters otherwise).

For all experiments comparing the coverage and interval width of JAWA to other influence function approximated baselines, we used a neural network predictor with one hidden layer consisting of 25 hidden units. Covariate and label data were centered and scaled. The neural network was trained for 2000 epochs with batch sizes of 50 and a learning rate of 0.0001, which generally resulted in convergence. The objective function for the neural network in JAWA is the negative log likelihood with a Gaussian prior or L2 regularization term. The L2 regularization was added to satisfy assumptions for computing IFs described in Giordano et al. [2019a] and due to empirical findings of first-order IFs for neural networks requiring regularization for reliable results [Basu et al., 2020]. The L2 regularization $\lambda$ parameter was tuned using a grid search prior to all experiments using a "tuning" validation set of 200 samples that were excluded from both the training and test sets in the experiments (see Appendix D.8 for more details regarding the L2 regularization tuning).

## D.3 Comparison of coverage variance for JAW and weighted split

Table 3: Coverage variance for JAW and weighted split conformal prediction, averaged across 1000 experimental replicates (i.e., statistics are the variance of all of the 1000 mean coverage statistics, one for each experiment). Lower coverage variances indicate more reliable coverage. The coverage variance for JAW is lower than that of weighted split conformal prediction in all datasets and predictor conditions due to JAW avoiding the sample splitting required by weighted split.

| Dataset | Airfoil | | Wine | | Wave | | Superconduct | | Communities | |
|---|---|---|---|---|---|---|---|---|---|---|
| Method | NN | RF | NN | RF | NN | RF | NN | RF | NN | RF |
| Weighted split | 0.0022 | 0.0023 | 0.0019 | 0.0017 | 0.0030 | 0.0029 | 0.0040 | 0.0035 | 0.00194 | 0.0021 |
| JAW | 0.0010 | 0.0019 | 0.0013 | 0.0015 | 0.0005 | 0.0014 | 0.0021 | 0.0030 | 0.00189 | 0.0014 |

## D.4 Additional AUC results

Due to space constraints, in the main paper Figure 5 we only report error assessment AUC results for the neural network predictor condition. For completeness, in Figure 8 we present error assessment results for both the neural network predictor (top row) and random forest predictor (bottom row), which are similar. Moreover, 8 also presents results for several baselines with reduced sample size to investigate how JAW's reduced effective sample size inherent to likelihood-ratio weighting may impact its performance. In particular, with dotted lines Figure 8 also presents the AUROC scores for jackknife+, CV+, and split conformal with the sample size for their predictive intervals reduced to 50 (note that only the sample size used to compute the predictive intervals was reduced to 50, not the sample size used to train $\widehat{\mu}$), because $\widehat{n} = 50$ is approximately the effective sample size of JAW in these experiments (see Appendix D.1). Relative to the methods with full sample size in the calculation of their predictive intervals, jackknife+, CV+, and split conformal prediction with reduced effective sample size have reduced AUROC scores. This suggests that JAW's AUROC is also likely negatively impacted by its reduced effective sample size, which could explain why JAW attains AUROC values comparable to jackknife+ despite holding the advantage over jackknife+ of coverage validity under covariate shift.

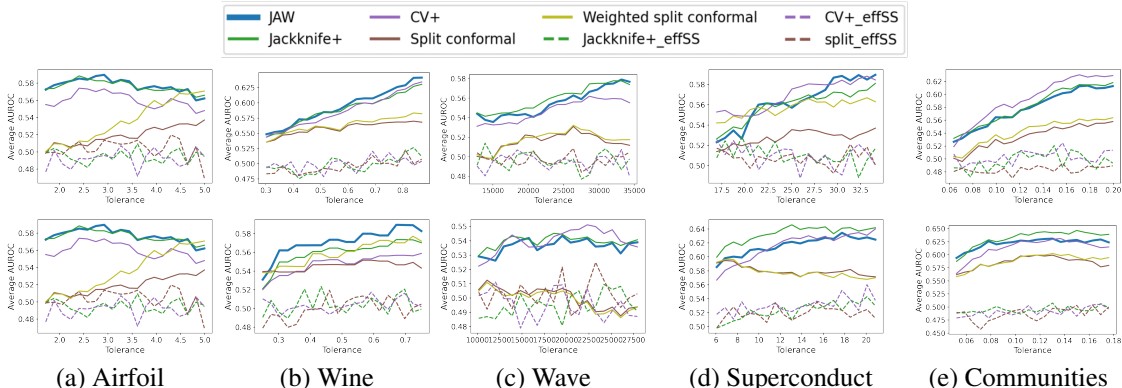

Figure 8: AUROC values for different tolerance levels $\tau$ across the three datasets, averaged across 50 experiment replicates, with neural network (top row) and random forest (bottom row) $\widehat{\mu}$ predictor. CV+-effSS, jackknife+-effSS, and split-effSS refer to the corresponding methods with sample size reduced to 50 for the construction of their predictive intervals, as described in Appendix D.4.

## D.5 JAW with estimated weights

JAW assumes access to oracle likelihood ratio weights, but that in practice this information is often not available. In such cases, the likelihood ratios can be estimated through an approach such as probabilistic classification, moment matching, or minimization of $\phi$-divergences (for a review of likelihood ratio estimation approaches see Sugiyama et al. [2012]). JAW's coverage performance will depend on the quality of the likelihood ratio estimates.

The following experiments compare coverage histograms of JAW with oracle likelihood-ratio weights those of JAW with weights estimated from probabilistic classification. We follow the approach used in Tibshirani et al. [2019] for estimating the likelihood-ratio weights using logistic regression and random forest classifiers. Specifically, for training covariate data $X_1, ..., X_n$ and test covariate data $X_{n+1}, ..., X_{n+m}$ where $C_i = 0$ for $i = 1, ..., n$ and $C_i = 1$ for $i = n+1, ..., n+m$, the conditional odds ratio $\mathbb{P}(C = 1|X = x)/\mathbb{P}(C = 0|X = x)$ can be used as an equivalent substitute to the likelihood ratio weight function $w(x)$ due to the normalization of the weights for use in JAW. Thus, for an estimate $\widehat{p}(x) \approx \mathbb{P}(C = 1|X = x)$ obtained from a classifier such as logistic regression or random forest, then we can use the following estimated weight function in place of likelihood-ratio weights:

$$\widehat{w}(x) = \frac{\widehat{p}(x)}{1 - \widehat{p}(x)}. \tag{47}$$

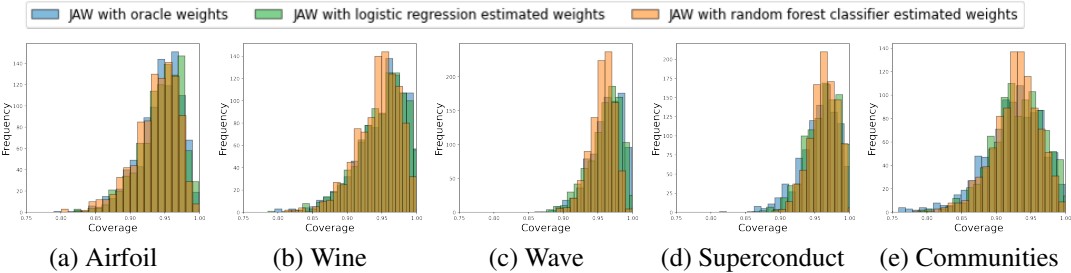

Figure 9: Comparison of JAW coverage under covariate shift with oracle versus estimated likelihood ratio weights for neural network predictor across all datasets. Blue is oracle weights, green is weights estimated with logistic regression, and orange is weights estimated with random forest classifier. Histograms represent 1000 experimental replicates.

Figure 9 illustrates the coverage performance of JAW with weights estimated by both logistic regression and random forest classifiers as described in Section D.2, compared to JAW with oracle weights. Results are for both neural network and random forest regression predictors across all five UCI datasets. We observe that the coverage histograms for JAW with both weight estimation methods are largely directly overlapping with the coverage histogram for JAW with oracle weights. These results demonstrate the applicability of JAW with estimated weights for predictive inference under covariate shift when the true likelihood ratio is not known but can be estimated from the data.

### D.6 Ablation studies on shift magnitudes

We demonstrate the effect of different magnitudes of covariate shift by comparing the coverage performance of JAW and the jackknife+ on the airfoil dataset with different magnitudes of the exponential tilting bias parameter $\beta$. Informed by these experiments depicted in Figure 10—where JAW's mean coverage remains consistent but the variance in coverage increases with increased covariate shift magnitude—we performed additional experiments to investigate the potential cause of JAW's increased variance. Specifically, we compare histograms of JAW's coverage at a fixed covariate shift magnitude to that of jackknife+ without covariate shift but with reduced "effective sample size", which is known to be reduced by likelihood ratio weighting. Tibshirani et al. [2019] made a similar comparison between weighted split conformal prediction under covariate shift and standard split conformal prediction with reduced effective sample size, and we use the same heuristic for effective sample size estimation [Gretton et al., 2009, Reddi et al., 2015] (which we also used for selecting exponential tilting parameter values for each dataset in Figure 3):

$$\widehat{n} = \frac{[\sum_{i=1}^{n} |w(X_i)|]^2}{\sum_{i=1}^{n} |w(X_i)|^2} = \frac{||w(X_{1:n})||_1^2}{||w(X_{1:n})||_2^2}.$$

**Effect of different magnitudes of covariate shift** As shown in Figure 10, the extent of covariate shift can be controlled by modifying a parameter in the exponential tilting weights so that weights are are more or less drastic. When the bias parameter is set to 0 this corresponds to no bias or IID train and test data. We can see that JAW is robust to different amounts of covariate shift, generating high coverage even under high level of shift.

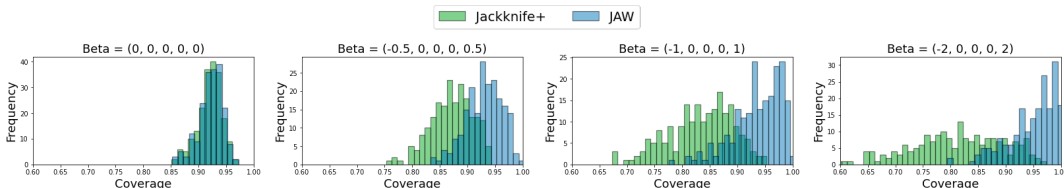

Figure 10: JAW performance compared to jackknife+ on the airfoil dataset with random forest $\widehat{\mu}$ function, under increasing magnitude of covariate shift (different $\beta$ values), with 200 replicates.

**Reduced effective sample size accounts for JAW increase in coverage variance under shift** While JAW's mean coverage remains relatively consistent under different magnitudes of covariate shift as seen in Figure 10, we also observe that the variance in coverage is higher for higher levels of shift. We hypothesized that this increase in variance is due to the high variance issue associated with important weighting methods that is well known [Reddi et al., 2015, Li et al., 2020] in the literature. We evaluate this hypothesis with effective sample size experiments reported in Figure 11 that compare a histogram of JAW's coverage under covariate shift with the coverage of jackknife+ with IID data but reduced effective sample size corresponding to the magnitude of covariate shift that JAW is evaluated on (see Appendices D.1 and D.6 for details). In Figure 11 we see that the coverage histogram for JAW under covariate shift is nearly directly overlapping with the histogram for jackknife+ coverage with no shift but reduced effective sample size. This result suggests that the reduction of effective sample size due to likelihood ratio weighting is largely if not entirely responsible for the increase in JAW coverage variance for increased shift magnitudes. We leave the variance reduction of our work to the future work.

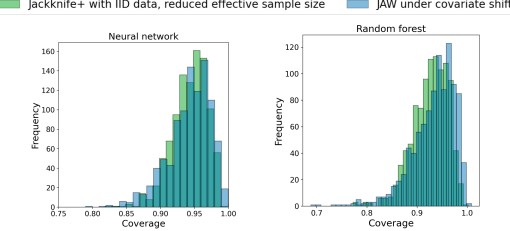

Figure 11: Comparison of JAW coverage histogram under covariate shift (blue) to jackknife+ coverage histogram (green) with no covariate shift but reduced effective sample size corresponding to the magnitude of covariate shift that JAW is evaluated on. Experiments are for both neural network (left) and random forest (right) predictors on the airfoil dataset, with 1000 experimental replicates. The largely overlapping histograms suggests that the increase in JAW coverage variance observed in Figure 10 is largely due to the decrease in effective sample size inherent to likelihood ratio weighting.

## D.7   Empirical runtime of JAWA compared to JAW

Whereas JAW requires retraining $n$ leave-one-out models, JAWA does not require any retraining, and thus generally enjoys significantly faster runtime than JAW. In Table 4 we report the empirical runtime of JAW compared to JAWA for different orders of JAWA's influence function approximation. In these experiments, JAWA is orders of magnitude faster than JAWA regardless of whether the influence function approximation is first, second, or third order (though of course the specific runtime statistics depend on the model architecture, optimization scheme, or dataset). JAWA's runtime does not increase substantially (relative to JAW's runtime) with increased influence function orders for $K \in \{1, 2, 3\}$.

Table 4: Example empirical comparison between the runtime for JAW and JAWA-$K$ for different influence function approximation orders $K \in \{1, 2, 3\}$ for the neural network predictor used in the JAWA experiments (see Appendix D.2), rounded up to the nearest second. This runtime experiment was performed on an 8-core personal computer with 32 GB of memory.

| Method | Airfoil | Wine | Wave | Superconduct | Communities |
|--------|---------|------|------|--------------|-------------|
| JAW | 58 min, 39 s | 59 min, 18 s | 1 hr, 24 min, 24 s | 1 hr, 26 min, 53 s | 1 hr, 25 min, 42 s |
| JAWA-1 | 1 s | 2 s | 4 s | 7 s | 8 s |
| JAWA-2 | 3 s | 4 s | 6 s | 11 s | 14 s |
| JAWA-3 | 11 s | 12 s | 16 s | 21 s | 23 s |

## D.8   L2 regularization for JAWA experiments

For the experiments involving JAWA and its baselines, the following L2 regularization tuning procedure was used for the neural network described in the second paragraph of D.2. The grid search evaluated the coverage of the first-order influence function approximation of the jackknife+ at different values of the regularization tuning parameter $\lambda \in \{0.5, 1, 2, 4, 8, 16, 32, 64, 96, 128\}$ for 10 train-test splits among all data for a dataset aside from the holdout tuning set. The smallest value of $\lambda$ in the grid search for which the coverage of the first-order influence function approximation of the jackknife+ exceeded 0.875 was used. The coverage calibration threshold of 0.875 was selected because the change in coverage due to increased $\lambda$ appeared to plateau just above or below the target coverage rate of 0.9 for each dataset, so setting the threshold slightly below 0.9 can help avoid over-regularizing. See Angelopoulos et al. [2020] for a discussion of calibrating uncertainty estimation in conformal prediction. This grid search procedure identified a separate $\lambda$ regularization parameter for each dataset: $\lambda_{\text{air}} = 1, \lambda_{\text{win}} = 8, \lambda_{\text{wav}} = 4, \lambda_{\text{sup}} = 96, \lambda_{\text{com}} = 64$. Additionally, we also added a dampening term to the Hessian (for IFs computation) as in Koh and Liang [2017] so that the smallest eigenvalue of the Hessian was at least 0.5.

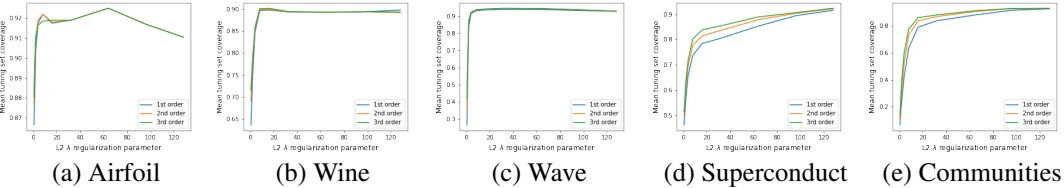

| (a) Airfoil | (b) Wine | (c) Wave | (d) Superconduct | (e) Communities |

Figure 12: Grid search plots for tuning the $\lambda$ L2 regularization parameter for influence function coverage experiments. All experiments are done with 1st, 2nd, and 3rd order influence function approximations of the jackkinfe+ (denoted in blue, orange, and green lines in the figure). The y-axis for each plot is the average coverage on the tuning dataset for each L2 regularization parameter $\lambda \in \{0.5, 1, 2, 4, 8, 16, 32, 64, 96, 128\}$.

### D.9 Histogram comparison of jackknife+ and JAW coverage under covariate shift

Figure 13 displays an example histogram comparison of jackknife+ and JAW coverage under covariate shift for both the neural network and random forest predictors on the airfoil dataset.

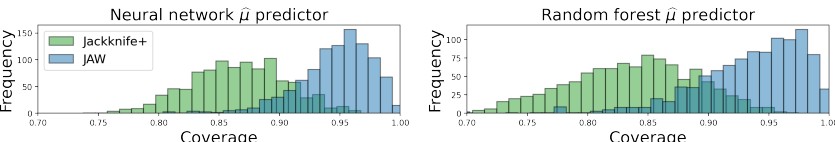

Figure 13: Jackknife+ versus JAW coverage under covariate shift for the airfoil dataset, when $\beta = (-1, 0, 0, 0, 1)$, for 1000 replicates. JAW maintains the high coverage under covariate shift.

### D.10 Cases where jackknife+ may not lose coverage

Although JAW maintains significantly higher coverage than jackknife+ in most conditions, our results suggest that there are some cases when jackknife+ may not lose coverage despite lacking a coverage guarantee for covariate shift. For instance, in Figure 3 jackknife+ does lose coverage for the random forest $\widehat{\mu}$ predictor, but it does not appear to lose coverage below the target level with the neural network $\widehat{\mu}$ predictor. Figure 14 allows for a closer look at this observation, with the coverage histograms for JAW and jackknife+ on the superconductivity dataset for both random forest and neural network $\widehat{\mu}$ predictors. In Figure 14 there does appear to be a slight loss of coverage for the jackknife+ with neural network $\widehat{\mu}$ predictor, but not as significant of a loss of coverage as with a random forest $\widehat{\mu}$.

A stronger example were jackknife+ appears to not lose coverage under covariate shift is the wave dataset, where JAW and jackknife+ appear to have similar coverage (Figure 3). Figure 15 examines this observation more closely by comparing JAW and jackknife+ coverage histograms corresponding to increasing levels of covariate shift. For the wave dataset, jackknife+ does not seem to lose coverage regardless of the extent of covariate shift.

Though we leave detailed analysis of the conditions that cause jackknife+ to lose coverage or not for future work, we conjecture that jackknife+ loss of coverage may be related covariate shift that makes difficult-to-predict datapoints more likely in the test distribution, and conversely that jackknife+ may not lose coverage when covariate shift does not make difficult-to-predict datapoints more likely in the test distribution. That is, the covariate shift method we use—exponential tilting—causes rare training points to be more common in the test distribution based on the $\beta$ used for tilting, but our conjecture is that the rarity of a datapoint in the training distribution does not necessarily determine how difficult that point is to predict. If rare but easy-to-predict datapoints are made more common due to exponential tilting, then this could explain why jackknife+ does not lose coverage in some cases as in Figure 15, though this conjecture requires further investigation.

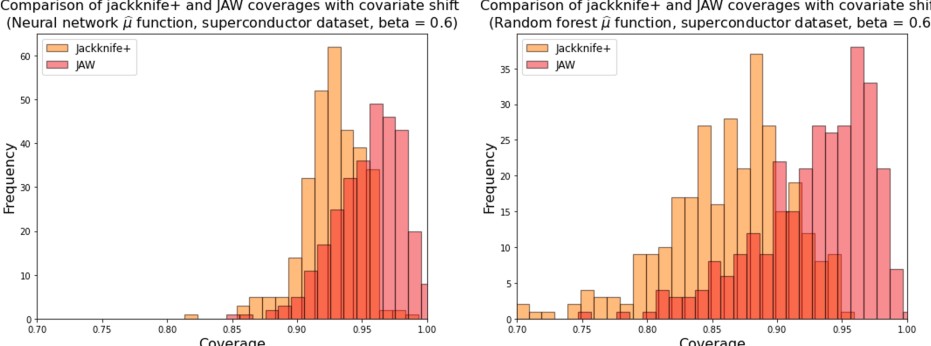

Figure 14: Comparison of the histogram of coverage on Superconductor dataset under covariate shift on the first principal component of the data, with tilting parameter $\beta = 0.6$. JAW still achieves high coverage while jackknife+ loses coverage significantly for the random forest $\widehat{\mu}$ predictor (right). For the neural network $\widehat{\mu}$ predictor (left), jackknife+ does not substantially lose coverage, while JAW has marginally higher coverage, illustrating minimal benefit of JAW over jackknife+ in this case. This is 300 replicates of the experiments.

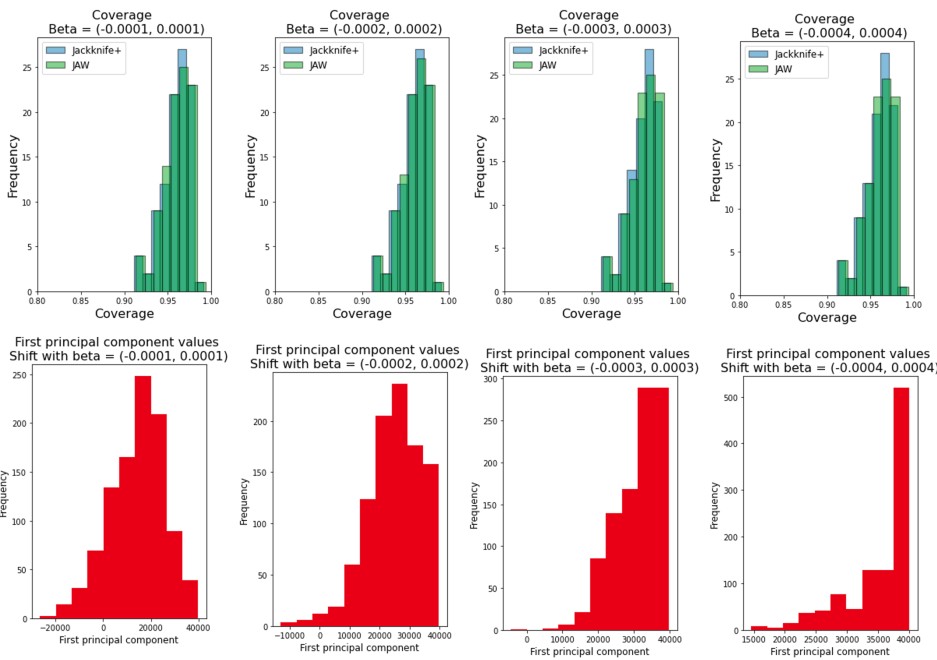

Figure 15: JAW and jackknife+ coverage for different levels of covariate shift levels on the wave energy converters dataset. Each column corresponds to a different level of shift, with increasing shift towards the right. The top row compares JAW (green) and jackknife+ (blue) coverage for a given shift level. The bottom row depicts the first principal component of the data at a given shift level. Neither jackknife+ nor JAW lose coverage at any tested shift level. This is 100 replicates of the experiments.

# E    Code and computational details

## E.1    Code:

https://github.com/drewprinster/jaws.git

## E.2   Computational details

Most experiments, aside from the runtime comparison described in Appendix D.7 Table 4 were performed on an institutional high performance computing cluster (HPC) using 10 CPUs with a total of 50GB of memory. Some experiments with the superconduct and communities datasets were run on the HPC with 20 CPUs and a total of 100GB of memory.