# OpenReview forum: "JAWS: Auditing Predictive Uncertainty Under Covariate Shift"
_NeurIPS.cc/2022/Conference — NeurIPS 2022 Accept_

### Official Review · Reviewer_AAyc · 2022-06-30

**Rating:** 4
**Confidence:** 3
**Soundness:** 3 good
**Presentation:** 3 good
**Contribution:** 3 good

**Summary:**

The authors focus on introducing a new method named JAW to estimate predictive inference under covariate shift, and it achieves superior performance over the baselines on different data sets and architectures of black-box model. The proposed method is a weighted version of the jackknife+, and the weights are not a new technique. What's more, the idea of a weighted version of conformal prediction under covariate shift is not new, so the innovation of this paper is limited.


**Questions:**

 1. The style of the citations is different for different sections.

  2. There are writing issues in line 107.

  3. There seems to be reference error in line 109 and line 110, line 168

**Limitations:**

See above

**Strengths And Weaknesses:**

Strength：
  The authors analyze the problem scenario and the motivation clearly.

Weaknesses：
  1. There seems to be an error in line 145, the model is trained with the ith datapoint removed, but the definition A(·) seems to be trained with the ith datapoint. And compared with \hat{\mu} defined in line 77, \hat{\mu_{-i}} seems have one more element, do authors mean \hat{\mu_{-i}(X_{n+1})}? Imprecise use of symbols makes definitions ambiguous.

  2. Many symbols are not formally defined when used. Although this is understandable due to space constraints, the authors had better add the note "as defined in [ ]" at least, or it will cause reading difficulties, and the reader needs to guess the meaning of each symbol according to the previous papers.

  3. The paper lacks innovation, the idea is based on [1] and the technique is from [1], then this technique is added to [2]. Hope the authors can put forward more analysis and improvement.


[1] Ryan J Tibshirani, et al. Conformal prediction under covariate shift. NeurIPS, 2019.

[2] Rina Foygel Barber, et al. Predictive inference with the jackknife+. The Annals of Statistics, 2021.

---

> ### Author Response · Authors · 2022-08-02
> **Response to reviewer AAyc**
>
> Thank you for your informative review!
>
> **Weaknesses 1, typo on line 145:**
> Apologies for this typo, we have fixed it in our revision. This line should have had $\mu_{-i} = A((X_1, Y_1), …, (X_{i-1}, Y_{i-1}), (X_{i+1}, Y_{i+1}), …, (X_n, Y_n))$
>
> **Weaknesses 2, formally defining symbols:**
> As best as possible we have taken care to screen the revision for typos, resolve any ambiguous notation, and overall improve clarity in our use of symbols. We would again review phrasing and notation with an eye to clarity
>
> **Weaknesses 3, innovation:**
> We refer to part 1b of our general reviewer response, where we emphasize how the JAW coverage guarantee does not follow trivially from applying the results from Tibshirani et al. 2019 to the jackknife+ in Barber et al. 2021, as well as highlight the other theoretical contributions we make that are not inspired by Tibshirani et al. 2019 nor Barber et al. 2021. Rather, our proof of Theorem 1 requires substantial generalization of the proof for the jackknife+ coverage guarantee (revision Section 3.1 and Appendix C.1 for details). Additionally, we highlight the originality and implications of our error detection / risk assessment guarantees (originally in our appendix, now Theorem 3 and Theorem 4 in our revision), which provide a general approach to repurposing any distribution-free predictive inference method and its guarantee to the risk assessment task (revision Section 3.3 and Appendix B.1).
>
> **Questions 1, citation styles:**
> In our revision we have made sure to standardize the citation style.
>
> **Questions 2, line 107:**
> In our revision we have done our best to eliminate any writing issues.
>
> **Questions 3, reference errors:**
> Thank you for pointing this out, we have resolved this reference error in our revision and done our best to catch and resolve any similar errors.

---

> > ### Author Response · Authors · 2022-08-08
> > **Reminder for any questions**
> >
> > Thank you again for taking the time to review our submission. We'd like to remind you that we're happy to answer any questions you may have before the author-reviewer discussion period ends tomorrow (August 9th).

---

> > > ### Comment · Reviewer_AAyc · 2022-08-09
> > > **Response to the author**
> > >
> > > Thanks for your response and most of my concerns have been addressed.

---

> > > > ### Author Response · Authors · 2022-08-09
> > > > **Thanks for acknowledging our response**
> > > >
> > > > Thanks for acknowledging our response! If there are no other outstanding concerns, we would really appreciate it if you’d consider updating your score. Thanks again for your helpful review, which helped us improve the paper.

---

> ### Comment · Area_Chair_8Z5m · 2022-08-08
> **Response needed**
>
> Dear Reviewer AAyc,
>
> Please kindly respond to the rebuttal provided by the authors and/or engage in the discussion with them. If it addresses your concerns, please react accordingly. Otherwise, please elaborate in your review on why you think the rebuttal/discussion is inadequate. Thank you.
>
> Best, AC

---

### Official Review · Reviewer_3P4m · 2022-07-05

**Rating:** 6
**Confidence:** 3
**Soundness:** 3 good
**Presentation:** 2 fair
**Contribution:** 2 fair

**Summary:**

Jackknife+ is an approach to construct a predictive interval with coverage guarantee under the exchangeable assumption. This paper extends Jackknife+ for covariate shift. In particular, the paper proves that the proposed Jackknife+ Weighted with data-dependent weight (JAW) predictive interval satisfies the same coverage guarantee as the original Jackknife+ given the likelihood ratio. The paper empirically demonstrated that the proposed JAW predictive interval satisfies a target coverage level, while generating a small interval width, compared to six baselines.


**Questions:**

Thanks for introducing an extension of Jackknife+ for covariate shift.

* As mentioned before my main concern lies in the practicality of Jackknife+ (though I believe that Jackknife+ could be useful) and the benefit of JAW compared to conformal prediction for covariate shift. In particular, Jackknife+ requires keeping n models to construct a predictive interval; in the case of a large dataset like ImageNet, which consists of 1.2M images, this means I need to keep 1.2M resnet152 to run Jackknife+, which does not sound feasible. In this sense, split conformal prediction is a significantly appealing approach, thus its extension to covariate shift (i.e., weighted split conformal prediction (WSCP) by Tibshirani et al. [2019]) would be enough for conformalized prediction intervals under covariate shift. In other words, the motivation to use JAW seems to be weak, though Jackknife+/JAW themselves have unique algorithmic features. Why or when can JAW be more useful than WSCP? If WSCP is less sample-efficient (as it needs a hold-out set), what is the range of the number of samples where JAW is better than WSCP? In practice, we anyway need a holdout set for model selection or likelihood ratio estimation, then why not properly exploit this holdout set along with WSCP for conformalized prediction sets?

* I believe that WSCP is also a good comparing approach, but it is missing in the main experiments, though SCP is included. I think a comparison to WSCP is needed to quantitatively understand the benefit of JAW as a conformalized prediction set under covariate shift.

* Barber et al. 2022 is clearly related work, but the proposed approach is not well contrasted to Babar et al. 2022. Why does Babar et al. 2022 make a stronger assumption (i.e., a fixed weights)? What are the main challenges on removing the stronger assumption? What's the key ideas on addressing the challenges?

**Limitations:**

The authors mentioned the high variance of coverage rate and computational overhead. In addition to this, the assumption that the likelihood ratio is known should be part of the limitations since otherwise JAW cannot get the desired guarantee.

**Strengths And Weaknesses:**

### originality
Compared to Barber et al 2022 (which is the most related work), this paper removes the strong assumption on the likelihood ratio (i.e., the likelihood ratio is fixed) to satisfy the coverage guarantee; this is novel, though it would be more instructive to state key differences on the generalization from Barber et al 2022. The proof technique seem to be a combination of Tibshirani et al. [2019] and Barber et al. [2021], but anyway this is a novel combination.

### quality
This paper may theoretically sound and the empirical results support the claim.

### clarity
Overall clearly written, but the following may enhance the readability.
* Barber et al 2022 also proposed the Jackknife+ extension for covariate shift, so it’s better to introduce it in Section 2 for clear comparison.
* line 145: I guess (X_{n+1}, Y_{n+1}) is a typo; the algorithm considers the point up to n.
* line 165: it contains an incomplete math notation.
* references to Appendix are broken.
* line 234: Jacknife-mm is not well defined; i is not a free-variable.

### significance
Extending uncertainty quantification approaches for covariate shift is required. But, the main concern lies in the usefulness of Jacknife+ in practical setups (i.e., it needs to keep n models to construct a predictive interval). So, extending Jackknife+ for covariate shift may not be a significant issue

---

> ### Author Response · Authors · 2022-08-02
> **Response to reviewer 3P4m (1 of 2)**
>
> Thank you for your informative review!
>
> **Clarity bullet 1, contrast to Barber et al. 2022:**
> We’ve responded to this point in our main reviewer response part 1a, and we’ve also addressed this concern in our revision Section 2.3 and Appendix A.3. In short, the key difference is that Barber et al. 2022 use fixed weights to compensate for unknown violations of the exchangeability (not limited to covariate shift) but at the expense of a bounded but generally nonzero “coverage gap” (drop in coverage relative to if the data were exchangeable), whereas our JAW method with data-dependent weights assumes covariate shift but does not suffer from any similar coverage gap. Moreover, the coverage guarantee in Barber et al. 2022 does not generalize to data-dependent likelihood ratio weights (the analysis from Barber et al. 2022 would give a trivial guarantee for JAW; Appendix A.3 of our revision has additional details).
>
> **Clarity bullet 2, typo on line 145:**
> Yes, this is a typo. Since line 145 refers to the leave-one-out model, this should be $\mu_{-i} = A((X_1, Y_1), …, (X_{i-1}, Y_{i-1}), (X_{i+1}, Y_{i+1}), …, (X_n, Y_n))$ (with a hat on the $\mu$, we couldn't add this in markdown). We’ve fixed this in our revision.
>
> **Clarity bullet 3, typo on line 165:**
> Our apologies, this is a typo. The math notation in this line should refer to the JAW predictive interval $C_{n, \alpha}^{\text{JAW}}(X_{n+1})$ (with a hat on $C$, could not add this in markdown). We’ve fixed this in our revision.
>
> **Clarity bullet 4, references to Appendix:**
> We’ve double checked that all the references, including those linking to the Appendix, are fixed in our revision.
>
> **Clarity bullet 5, jackknife-mm definition:**
> Thank you for catching this typo--each of the $\widehat{\mu}$ in line 234 should have been $\widehat{\mu}_{-i}$. We’ve fixed this in our revision, and we’ve taken extra care to screen for similar errors.
>
> **Questions bullet 1, practicality of JAW:**
> We agree that computational scalability is a key limitation to jackknife+ based methods like JAW. We’ve responded to this concern in our general reviewer responses, part 2, where we propose computationally efficient approximations to JAW using higher-order influence functions, which approach the JAW guarantee in the limit of sample size or influence function order, under regularity conditions now described in Section 3.2 of our revision.
>
> Additional details on JAWA computation: Computing the Kth order JAWA requires that the original predictor’s parameters are a local minimum of the objective function in strongly convex local region of the objective (that the Hessian is strongly positive definite), as well as the existence of the K+1th directional derivatives of the objective function at the local solution. (In practice, we can tune the L2 regularization of the model and add a dampening term to the Hessian in the IF computations to ensure that the requirements for the local convexity of the objective / Hessian positive definiteness are met--see our revision Appendix A.4.) JAWA then trades JAW’s computational requirement of retraining n models for the generally much milder computational requirement of computing and inverting the Hessian of the objective function and computing 1st through K+1th order directional derivatives of the objective. To compute JAWA influence functions we implement the recursive procedure from Giordano et al. 2019a, which leverages memory-efficient forward automatic differentiation and only requires computing and inverting the Hessian once for all influence function orders.
>
> JAWA coverage results: Section 4.3.2 and Figure 5 in our revision report coverage results for JAWA relative to baselines. From these experiments we see that across three of five datasets (airfoil, wine, and communities) JAWA is the only method that consistently reaches or nearly reaches the target coverage level for all tested influence function orders. In the remaining two datasets, either all baselines reach the target coverage along with JAWA (wave dataset) or JAWA outperforms IF-jackknife and IF-jackknife+ (superconduct dataset). For more analysis and discussion, please see our revision Sections 2.5 and 3.2, as well as Appendix A.4. In our revision, we also proof that JAWA approaches the JAW coverage guarantee in the limit of either the training data sample size or the influence function order (given Assumptions 1-4 and either Condition 2 or Condition 4 in Giordano et al. 2019a). In an example of empirical runtime comparison, JAWA can offer orders of magnitude faster computation time compared to JAW (revision Appendix D.7).

---

> > ### Author Response · Authors · 2022-08-02
> > **Response to reviewer 3P4m (2 of 2)**
> >
> > **Questions bullet 1, When JAW is more useful than WSCP:**
> > JAW generally holds the statistical advantage over WSCP of maintaining the accuracy of the full model by avoiding sample splitting, at the cost of JAW’s more demanding computational requirements. Specifically, the reduced sample size of WSCP relative to JAW results in an increased variance in coverage probability, which can be exacerbated in the covariate shift scenario where the likelihood ratio weights used in both WSCP and JAW already reduce each method’s effective sample size. In response to your point we have implemented WSCP as an additional baseline, and the following table compares the coverage variance for WSCP compared with JAW (also presented in our revision Appendix D.3). In all conditions, the coverage variance is higher for WSCP than for JAW (meaning that the coverage from WSCP is less reliable than for JAW).
> >
> > **Table comparing coverage variance of weighted split and JAW:**
> > | Dataset        | Airfoil | Airfoil | Wine   | Wine   | Wave   | Wave   | Superconduct | Superconduct | Communities | Communities |
> > |----------------|---------|---------|--------|--------|--------|--------|--------------|--------------|-------------|-------------|
> > | Predictor      | NN      | RF      | NN     | RF     | NN     | RF     | NN           | RF           | NN          | RF          |
> > | Weighted split | 0.0022  | 0.0023  | 0.0019 | 0.0017 | 0.0030 | 0.0029 | 0.0040       | 0.0035       | 0.00194     | 0.0021      |
> > | JAW            | 0.0010  | 0.0019  | 0.0013 | 0.0015 | 0.0005 | 0.0014 | 0.0021       | 0.0030       | 0.00189     | 0.0014      |
> >
> > **Questions bullet 1, repurposing a holdout set used for another purpose for WSCP:**
> > First, using a holdout set for model selection (e.g., hyperparmaeter tuning) would invalidate the holdout set for WSCP (or split conformal) because such a procedure would mean that the model-fitting algorithm $\mathcal{A}$ no longer treats the holdout data symmetrically relative to the test data. That is, the fact that holdout points are used to tune hyperparameters (or otherwise for model selection) whereas the test points do not alter hyperparameters breaks this assumed symmetry, making the holdout and test data no longer (weighted) exchangeable for the sake of WSCP. Next, a holdout set used for likelihood ratio estimation could potentially be used for WSCP assuming that the weight estimation procedure is independent of the model fitting and model selection. However, in practice many likelihood ratio estimation techniques in the covariate shift do not require a holdout set because access to the training and test data covariates is sufficient for this task. For instance, a probabilistic classification approach to likelihood ratio estimation only requires the training and test data covariates and class indications of whether each point is from the training data or the test data. Thus, in either the case of a holdout set used for model selection or for likelihood ratio estimation, relative to JAW WSCP would still have stronger requirements for sample splitting.
> >
> > **Questions bullet 1, WSCP as a comparison approach:**
> > As mentioned in the previous response, in response to your point we’ve implemented WSCP and included it as an additional baseline in our revision. WSCP and JAW achieve similar mean coverage across datasets and predictor functions, but the variance in the WSCP coverage is generally higher than that of JAW (due to sample splitting in WSCP); in our experiments, the variance in WSCP is higher than that of JAW for all 10 experimental conditions (5 datasets x 2 predictor types).
> >
> > **Questions bullet 2, comparison to Barber et al. 2022:**
> > We have responded to this point in Part 1 of our general response. Also see our response to Clarity bullet 1 (contrast to Barber et al. 2022).
> >
> > **Limitations: Assumption that likelihood ratio is known**
> > This limitation can be addressed in practice by estimating the likelihood ratio weights using techniques such as probabilistic classification, moment matching, or minimization of $\phi$-divergences (for a review of likelihood ratio estimation approaches see Sugiyama et al. 2012). We implement experiments of JAW with estimated weights (JAW-E), with both logistic regression and random forest classifiers for the weight estimation. JAW-E achieves similar coverage performance relative to JAW with oracle weights across all five UCI datasets and for both neural network and random forest predictors. For more details, please see our revision Appendix D.5.

---

> > > ### Comment · Reviewer_3P4m · 2022-08-05
> > > **Thanks for the response!**
> > >
> > > Thank you for the response along with additional experiments. The response addresses my concerns and I have updated my score due to the following reasons:
> > >
> > > * JAW is computationally expensive, but this issue is addressed by introducing higher-order influence functions (IF, proposed by Giordano et al. 2019); even though using IF with jackknife is not new, I would put more value on demonstrating the efficacy and efficiency of IF along with JAW (i.e., JAWA) over real datasets.
> > > * JAWA is asymptotically valid, but Figure 4 empirically justifies JAWA still satisfies the coverage rate.
> > > * I would say WSCP is superior to JAW as Figure 3 shows that WSCP and JAW achieve similar coverage rate but computational efficiency is more important than the high variance of coverage; mild high variance of WSCP won't matter if JAW takes 1 hour to run. However, JAWA addresses this computational issue. Now, I think WSCP is as good as JAWA as each of which has different limitations (i.e., high variance v.s. asymptotic guarantee).
> > > * I believe that JAW and JAWA enrich the diversity of conformalized predictors.
> > >
> > > Thanks for your contributions!

---

> > > > ### Author Response · Authors · 2022-08-07
> > > > **Thank you for your consideration!**
> > > >
> > > > Thank you very much for your consideration! We appreciate your feedback, which helped inform our additional experiments and improvements.

---

### Official Review · Reviewer_qDpq · 2022-07-10

**Rating:** 7
**Confidence:** 3
**Soundness:** 4 excellent
**Presentation:** 4 excellent
**Contribution:** 3 good

**Summary:**

This work introduces a new method, JAW, that incorporates importance weighting into Jackknife+. They demonstrate that JAW extends Jackknife+’s coverage guarantee to situations with covariate shift. Finally, they demonstrate the effectiveness of this method on experiments.

**Questions:**

* In line 144-145: $\hat{\mu}\_{-i}$ includes $(X\_{n+1}, Y\_{n+1})$, but that doesn’t seem to be the case for the definition of $\hat{\mu}_{-i}$ in [1]. Are these supposed to be (slightly) different things?
* I may have missed this, but what are the values of $\beta$ for Figs. 3 and 4?
* Also regarding Fig 3., it seems that JAW and Jackknife-mm have comparable performance when using random forests, while Jackknife-mm can have much larger interval widths with neural networks. Is there an intuitive reason for this?
* Why are Naive, Jackknife-mm, and Jackknife missing from Fig. 4? Particularly Jackknife-mm as it seemed like the most competitive baseline in Fig. 3.

[1] Barber, Rina Foygel, et al. "Predictive inference with the jackknife+." The Annals of Statistics 49.1 (2021): 486-507.

**Limitations:**

* The main limitations of this method are adequately discussed in the work: in particular, the computational difficulty and the variance of JAW.
* This method, like other IW-based methods, requires knowing the covariate shift from train to test, which may not be applicable in many settings. Is there a way to incorporate weight estimates?

**Strengths And Weaknesses:**

Strengths:
* An valuable work in developing the Jackknife+ framework and extending its utility to an important use case.
* Good technical contribution

Weaknesses:
* The method so far seems infeasible for larger-scale problems

---

> ### Author Response · Authors · 2022-08-02
> **Response to reviewer qDpq (1 of 2)**
>
> Thank you for your informative review!
>
> **Questions: typos in lines 144-145:**
> ​​Thank you for pointing out this typo, this line should have had $\widehat{\mu}_{-i} = \mathcal{A}\Big((X_1, Y_1), …, (X_{i-1}, Y_{i-1}), (X_{i+1}, Y_{i+1}), …, (X_n, Y_n)\Big)$. We have fixed this typo in our revision.
>
> **Questions: Values of beta:**
> The details for these parameters and generally regarding the creation of covariate shift can be found in our revision Appendix Section D.1. In our revision we have standardized these tilting parameters in our main experiments so that for each dataset the resulting covariate shift results in a reduction of the effective sample size of the training data from 200 to approximately 50.
>
> **Questions: Why jackknife-mm seems to have larger interval width than JAW on neural networks more so than random forest:**
> Whereas JAW constructs predictive intervals using weighted quantiles on the $\mu_{-i}(X_{n+1}) \pm R_i^{LOO}$ values, jackknife-mm uses the minimum and maximum values of $\mu_{-i}(X_{n+1})$ and unweighted quantiles on $R_i^{LOO}$ in its interval construction. The jackknife-mm’s use of the min and max $\mu_{-i}(X_{n+1})$ values is the aspect of the method that gives it strictly larger intervals than the jackknife+, and is thus responsible for your observation of large jackknife-mm interval widths with the neural network predictor. In particular, these min and max values are highly sensitive to the presence of highly influential training points (i.e., if $i$ is an influential point then $|\mu_{-i}(X_{n+1}) - \mu(X_{n+1})|$ will be large), so jackknife-mm intervals will be especially large relative to JAW when there are few training point(s) that have an abnormally large effect on the prediction. Because neural networks are often more susceptible to overfitting than (ensemble) random forest predictors, we suspect that the neural network we use in our experiments is more likely to have high variability in its leave-one-out predictions relative to random forest. (Note: In this comment each $\mu$ should have a hat on it, we couldn't add this formatting in markdown.)
>
> **Questions: Why naive, jackknife-mm, and jackknife baselines were missing in AUC experiments:**
> The naïve and jackknife baselines were excluded from the figure for the second set of experiments because jackknife+ is generally superior to both the naïve and classic jackknife methods and to improve the readability of the figure (i.e., reducing the number of overlapping lines to interpret). The jackknife-mm method was excluded because its over-conservative construction makes it much less suitable to the error detection task (or “risk assessment” task, as we refer to the task in our revision), though perhaps we could add it to the final figure version to emphasize this point. To understand why jackknife-mm is not well suited to risk assessment, recall (or see Section 3.3 and Appendix B.1 where we elaborate) that our general approach to adapting a predictive inference method to risk assessment (assessing the “risk” or probability that the true label lies outside a margin-of-error interval around the full model prediction) requires identifying a predictive interval that is a sub-interval of the user-specified margin-of-error interval. For all other predictive inference baselines, the size of a predictive interval can be finely tuned by adjusting the target miscoverage level $\alpha$ that dictates the quantile levels on the predictive distribution, with large values of $\alpha$ corresponding to smaller intervals. However, even for a very large value of $\alpha$ the jackknife-mm’s predictive interval will be very large relative to that of other predictive inference baselines (due to the method’s use of the minimum and maximum $\mu_{-i}(X_{n+1})$ values), meaning that in many cases of our AUC experiments the jackknife-mm would not have a predictive interval that is a sub-interval of the user-specified margin-of-error interval, and thus would predict 0 probability in all of these cases. (Note: In this comment the $\mu$ should have a hat on it, we couldn't add this formatting in markdown.)

---

> > ### Comment · Reviewer_qDpq · 2022-08-06
> > **Response to authors**
> >
> > Thanks for the responses! My questions were well addressed and the restructuring of the paper and additional experiments seem good (though the figures could probably be a bit bigger). I have updated my score accordingly.

---

> > > ### Author Response · Authors · 2022-08-07
> > > **Thank you for your consideration!**
> > >
> > > Thank you very much for your consideration! In a final version of the paper we would do our best to make the figures a bit larger. We appreciate your feedback, as it helped inform our additional experiments and improvements.

---

> ### Author Response · Authors · 2022-08-02
> **Response to reviewer qDpq (2 of 2)**
>
> **Limitations, bullet 1: Computational difficulty**
> We agree that computational scalability is a key limitation to jackknife+-based methods like JAW. We’ve responded to this concern in our general reviewer responses, part 2, where we propose computationally efficient approximations to JAW using higher-order influence functions, which approach the JAW guarantee in the limit of sample size or influence function order, under regularity conditions now described in Section 3.2 of our revision.
>
> JAWA coverage results: In our revision, Figure 5 and Section 4.3.2 report coverage results for JAWA relative to baselines. From these experiments we see that across three of five datasets (airfoil, wine, and communities) JAWA is the only method that consistently reaches or nearly reaches the target coverage level for all tested influence function orders. In the remaining two datasets, either all baselines reach the target coverage along with JAWA (wave) or JAWA outperforms IF-jackknife and IF-jackknife+ (superconduct). In our revision, we also prove that JAWA approaches the JAW coverage guarantee in the limit of either the training data sample size or the influence function order (given Assumptions 1-4 and either Condition 2 or Condition 4 in Giordano et al. 2019a). In an example of empirical runtime comparison, JAWA can offer orders of magnitude faster computation time compared to JAW (revision Appendix D.7).
>
> **Questions, bullet 2: incorporating weight estimation:**
>
> This limitation can be addressed in practice by estimating the likelihood ratio weights using techniques such as probabilistic classification, moment matching, or minimization of $\phi$-divergences (for a review of likelihood ratio estimation approaches see Sugiyama et al. 2012). We implement experiments of JAW with estimated weights (JAW-E), with both logistic regression and random forest classifiers for the weight estimation. JAW-E achieves similar coverage performance relative to JAW with oracle weights across all five UCI datasets and for both neural network and random forest predictors. For more details, please see our revision Appendix D.5.

---

### Official Review · Reviewer_MNhq · 2022-07-12

**Rating:** 5
**Confidence:** 3
**Soundness:** 3 good
**Presentation:** 3 good
**Contribution:** 2 fair

**Summary:**

This paper studies conformal prediction under covariate shift and presents a weighted version of jackknife+ for solving it.

**Questions:**

- The notations in the paper seem not consistent. For example, the quantile functions in Eq. (4) and Eq. (6) use different notations.
- Is the equation of likelihood ratio correct in Line 105, Page 3? Does X_i follow the training distribution or the test distribution?

**Limitations:**

None.

**Strengths And Weaknesses:**

Strengths:
- The problem this paper studies is significant.
- Overall, the paper is well written.
- The theoretical analysis looks solid.

Weaknesses:
- The major problem of this paper is its originality. The authors claim that the weighted version of jackknife+ proposed in [Barber et al., 2022] uses fixed and not data-dependent weights (and thus differs from the authors' proposed weighted jackknife+). However, in my understanding, Barber et al. (2022) consider conformal prediction in a more general setting (non-exchangeable), and therefore, the weights in their method are not specified. More importantly, I think the weights should reduce to those appearing in the proposed method of this paper if we consider the covariate shift setting (a special case of non-exchangeability), which implies the proposed method of this paper can be regarded as a special case of [Barber et al., 2022]. And obtaining these weights under covariate shift looks not difficult as long as we follow [Tibshirani et al., 2019] and [Barber et al., 2022].

---

> ### Author Response · Authors · 2022-08-02
> **Response to reviewer MNhq**
>
> Thank you for your informative review!
>
> **Weaknesses: Originality relative to Barber et al. 2022**
> We have addressed this concern in part 1 of our general response. But briefly, in our general response part 1a and now also in our revision Section 2.3 and Appendix A.3, we clarify that JAW is not a special case of the approach proposed in Barber et al. 2022: Barber et al. 2022 compensate for unknown violations of the exchangeability (not limited to covariate shift) but at the expense of a bounded but generally nonzero “coverage gap”, whereas our JAW method with data-dependent weights assumes covariate shift but does not suffer from any similar coverage gap. Moreover, if one were to try to define JAW as a special case of the approach in Barber et al. 2022 (Section 5.3) with data-dependent likelihood ratio weights, then the analysis from Barber et al. 2022 would generally only give a trivial coverage guarantee for JAW (promising probability >=0).
> Additionally, in our general response part 1b we emphasize that the JAW coverage guarantee does not follow trivially from applying the results from Tibshirani et al., 2019 to the jackknife+ in Barber et al. 2021, as well as highlight the other theoretical contributions we make that are not inspired by Tibshirani et al., 2019 nor Barber et al. 2021. Specifically, our proof of Theorem 1 requires substantial generalization of the proof for the jackknife+ coverage guarantee (revision Section 3.1 and Appendix C.1 for details). Additionally, we highlight the originality and implications of our error detection / risk assessment guarantees (originally in our appendix, now Theorem 3 and Theorem 4 in our revision), which provide a general approach to repurposing any distribution-free predictive inference method and its guarantee to the risk assessment task (revision Section 3.3 and Appendix B.1).
>
> **Questions, bullet 1: Consistency in notation:**
>
> We apologize for any confusion that this may have caused. In our revision we have taken care to standardize our notation for improved consistency and clarity.
>
> **Questions: Correctness of line 105:**
> We can confirm that the notation is correct at this location. X_i follows the training distribution for $i \in \{1, …, n\}$ and it follows the test distribution if $i = n+1$. The likelihood ratio is defined for both cases, i.e., regardless of whether $X_i$ is from the training or test distribution.

---

> > ### Author Response · Authors · 2022-08-08
> > **Reminder for any questions**
> >
> > Thank you again for taking the time to review our submission. We'd like to remind you that we're happy to answer any questions you may have before the author-reviewer discussion period ends tomorrow (August 9th).

---

> ### Comment · Area_Chair_8Z5m · 2022-08-08
> **Response needed**
>
> Dear Reviewer MNhq,
>
> Please kindly respond to the rebuttal provided by the authors and/or engage in the discussion with them. If it addresses your concerns, please react accordingly. Otherwise, please elaborate in your review on why you think the rebuttal/discussion is inadequate. Thank you.
>
> Best, AC

---

### Author Response · Authors · 2022-08-02
**General response to reviewers’ shared concerns**

We thank the reviewers for their thoughtful and informative feedback! In this general response we will address points that were shared by multiple reviewers and summarize the changes that were made in our revision. We will address individual points in greater detail in the reviewer-specific responses.

**(1) Novelty / originality:**

**(1a) Novelty / originality relative to Barber et al. 2022 (3P4m and MNhq):** We would like to clarify how our proposed approach is distinct from and is not a special case of Barber et al. 2022. In short, the key difference is that Barber et al. 2022 use fixed weights to compensate for unknown violations of the exchangeability (not limited to covariate shift) but at the expense of a bounded but generally nonzero “coverage gap” (drop in guaranteed probability that the predictive interval contains the true label compared to coverage guarantee if the data were exchangeable), whereas our JAW method with data-dependent weights assumes covariate shift but does not suffer from any similar coverage gap. Moreover, the coverage guarantee in Barber et al. 2022 does not generalize to data-dependent likelihood ratio weights (the analysis from Barber et al. 2022 would give a trivial guarantee for JAW). For more details, please see our revised supplement, Appendix A.3.

**(1b) Novelty / originality relative to Tibshirani et al. 2019 and Barber et al. 2021 (AAyc):** For clarification, we would like to emphasize that the JAW coverage guarantee that we present in our Theorem 1 does not simply follow from applying the results of Tibshirani et al. 2019 to the jackknife+ method in Barber et al. 2021. That is, our proof of Theorem 1 (in revised supplement, Appendix C.1) requires substantial generalization of the proof for the jackknife+ coverage guarantee. We refer to our Remark in our revision, Section 3.1 and Appendix C.1 for more details. Additionally, we would also like to highlight the implications of our error detection / risk assessment guarantees, which are not inspired by Tibshirani et al. 2019 nor Barber et al. 2021 (these results were initially included in our supplement, but we have updated them as Theorem 3 and 4 in our revision for improved clarity). Whereas related works focus on the task of generating a predictive interval that satisfies a user-specified coverage level, these results present a general approach to adapting any predictive inference method (e.g., conformal prediction, jackknife+) and associated guarantee to the converse task of estimating the coverage probability for a user-specified interval, such as defined by a margin of acceptable error around a model’s prediction. This “risk assessment” task is applicable to safety-critical scenarios with a strict margin of error (e.g., cancer therapy dose prediction). For further motivation we refer to our revision’s Appendix A.1, and for further details we refer to our revision’s Section 3.3 and Appendix B.1.

**(2) Computational scalability of JAW (3P4m and qDpq):** To address JAW’s computational limitation, we have implemented, performed additional experiments with, and present asymptotic analysis for a sequence of JAW Approximations (which we call JAWA) that use higher-order influence functions to avoid the computationally costly retraining required by JAW. Influence functions allow us to approximate how a predictor’s parameters will change in response to reweighting (or in our case, removing) a training point, and thus they enable us to approximate the leave-one-out model parameters that JAW requires via a (potentially higher order) Taylor expansion of the model parameters with respect to data weights. We omit a more comprehensive discussion of influence functions in this response for the sake of brevity, but for more details regarding influence functions background and our implementation please see our revision Section 2.5 and Appendix A.4. In our experiments, JAWA achieves the target coverage level for most datasets and influence function orders whereas baselines often lose coverage (revision Section 4.3.2 and Figure 5). The following table provides an example empirical comparison of JAW versus JAWA runtime for different orders of the JAWA influence function approximation (experimental details in our revision Appendix D.7 and D.8):

| Method | Airfoil      | Wine         | Wave               | Superconduct       | Communities        |
|----|-------|-------|----------|----------|---------|
| JAW    | 58 min, 39 s | 59 min, 18 s | 1 hr, 24 min, 24 s | 1 hr, 26 min, 53 s | 1 hr, 25 min, 42 s |
| JAWA-1 | 1 s          | 2 s          | 4 s                | 7 s                | 8 s                |
| JAWA-2 | 3 s          | 4 s          | 6 s                | 11 s               | 14 s               |
| JAWA-3 | 11 s         | 12 s         | 16 s               | 21 s               | 23 s               |

---

### Author Response · Authors · 2022-08-02
**Summary of changes in the revision**

We have uploaded a revision of our paper and our supplement, and we would like to highlight the following important changes that we made in our revision in response to reviewers’ comments:

- Clarified comparison to Barber et al. 2022 (Part 1a of general response)

- Reorganization and emphasis of risk assessment guarantees (Part 1b of general response): We have moved our error detection guarantees from the appendix to the main paper (revision, Theorem 3a and 3b) and we have reframed the guarantees as for the “risk assessment” task for improved clarity.

- JAWA (Part 2 of general response): We’ve added experiments and analysis of a computationally efficient approximation to JAW using higher-order influence functions.

- Additional experimental comparison baseline (3P4m): We have added weighted split conformal prediction as an additional baseline for comparison to JAW (revision Section 4.3.1 and Appendix D.3). JAW and weighted split achieve similar coverage, but due to sample splitting weighted split has higher coverage variance than JAW in all conditions (Appendix D.3).

- Additional datasets: We have added additional UCI datasets for all experiments, for now a total of 5 datasets. JAW achieves target coverage in all datasets, along with the jackknife-mm and weighted split baselines (revision Section 4.3.1 and Figure 4); however in all conditions jackknife-mm’s intervals are wider (and thus less informative) than JAW’s and due to sample splitting weighted split has higher coverage variance (and thus less reliable coverage) than JAW (revision Appendix D.3). JAW also performs as well or better than baselines on the AUC risk assessment experiments for all datasets (revision Section 4.3.3 and Figure 6).

- Supplementary experiments: To the appendix, we have added coverage experiments with estimated likelihood ratio weights (in response to 3P4m and qDpq; Appendix D.5), which achieve similar empirical performance as oracle weights, as well as ablation studies that demonstrate JAW’s higher coverage variance under covariate shift is due to reduced effective sample size that is known to be a result of importance weighting methods (Appendix D.6).

- Clarity and notation: We have reorganized and revised some sections for clarity and consistent notation.

Note: On August 5, 2022 we submitted a second revision with a 9 page main paper after realizing that on August 2, 2022 we had been confused about the revision page limit when we had submitted a 10 page main paper. We apologize for any inconvenience.

---

### Meta-Review · Area_Chair_8Z5m · 2022-08-28

**Recommendation:** Accept
**Confidence:** Less certain

**Metareview:**

This paper studies conformal prediction under covariate shift and proposes JAW, a weighted version of jackknife+ to solve this problem. There is a consensus among the expert reviewers that the paper considers an important problem and has substantial contributions that are deemed adequate for publication at NeurIPS2022. The authors provided a rebuttal that has sufficiently addressed the reviewers' concerns as acknowledged by Reviewers `qDpq `, `3P4m`, and `AAyc`.


**Award:**

No

---

### Decision · Program_Chairs · 2022-09-14

Accept